# Continuous Partitioning for Graph-Based Semi-Supervised Learning

**Chester Holtz**
Halicioğlu Data Science Institute
University of California San Diego
La Jolla, CA
chholtz@ucsd.edu

**Pengwen Chen**
Department of Applied Mathematics
National Chung Hsing University
South District, Taichung, Taiwan
pengwen@email.nchu.edu.tw

**Zhengchao Wan**
Department of Mathematics
University of Missouri
Columbia, MO
zwan@missouri.edu

**Chung-Kuan Cheng**
Department of Computer Science
University of California San Diego
La Jolla, CA
ckcheng@ucsd.edu

**Gal Mishne**
Halicioğlu Data Science Institute
University of California San Diego
La Jolla, CA
gmishne@ucsd.edu

## Abstract

Laplace learning algorithms for graph-based semi-supervised learning have been shown to suffer from degeneracy at low label rates and in imbalanced class regimes. Here, we propose CutSSL: a framework for graph-based semi-supervised learning based on continuous nonconvex quadratic programming, which provably obtains *integer* solutions. Our framework is naturally motivated by an *exact* quadratic relaxation of a cardinality-constrained minimum-cut graph partitioning problem. Furthermore, we show our formulation is related to an optimization problem whose approximate solution is the mean-shifted Laplace learning heuristic, thus providing new insight into the performance of this heuristic. We demonstrate that CutSSL significantly surpasses the current state-of-the-art on $k$-nearest neighbor graphs and large real-world graph benchmarks across a variety of label rates, class imbalance, and label imbalance regimes. Our implementation is available on github[1].

## 1 Introduction

In semi-supervised learning (SSL), a learner is given access to a partially-labeled training set consisting of labeled examples and unlabeled examples. The goal in this setting is to learn a predictor that is superior to a predictor that is trained using the labeled examples alone. This framework is motivated by the high cost of obtaining annotated data on practical problems. For problems where very few labels are available, the geometry of the unlabeled data can be used to significantly improve the performance of classic machine learning models. A seminal work in graph-based semi-supervised learning is Laplace learning [38], which seeks a harmonic function that extends provided labels over

---

[1] https://github.com/Mishne-Lab/CutSSL

38th Conference on Neural Information Processing Systems (NeurIPS 2024).

the unlabeled vertices (a *harmonic extension* of the labeled vertices). Laplace learning and its variants have been widely applied in semi-supervised and graph-structured learning [37, 36, 2, 20].

However, recent work [26, 1] has demonstrated that 'vanilla' Laplace learning methods exhibit poor prediction error at low label rates, primarily due to degenerate estimates near the decision boundary—although this can be partially addressed in practice via a simple mean-shift of the predictions [10]. Furthermore, in the context of classification, the labels often correspond to elements from a discrete set. In this setting, typically a thresholding operation is applied post hoc to the harmonic extension to map from continuous predictions to the discrete set of labels. This thresholding can further exacerbate the aforementioned degeneracy. Furthermore, when class sizes are imbalanced, approximate heuristics are often employed to ensure satisfaction of the volume (class-cardinality) prior [10, 19]. In particular, [19] apply efficient auction algorithms [5] to volume-constrained semi-supervised learning. They do this by iteratively solving linearizations of a quadratic problem under volume and box constraints to reach a local solution. The authors claim convergence to a local solution due to monotonic decrease of the objective, although local solutions may be poor and convergence can be slow.

In this paper, we formulate cardinality-constrained semi-supervised learning as a nonconvex quadratic program, a generalization of the assignment problem. Notably, in contrast to semidefinite or spectral relaxations, our formulation is exact in that it has a $0 - 1$ minimizer corresponding to a maximally smooth discrete label assignment. Furthermore, our method has connections to spectral graph partitioning and the mean-shift Laplace learning heuristic.

Our method involves solving a sequence of quadratic programs via Alternating Direction Method of Multipliers (ADMM) [7], which exhibits robust convergence guarantees, having also been applied successfully to Quadratic Assignment Problems and similar Semi-Definite Program (SDP)-based relaxations of nonconvex Boolean constrained problems [24]. As we will show, the ADMM iterates can be solved efficiently, particularly when the underlying graph is sparsely connected.

Our contributions are as follows:

1. We introduce a framework derived from an exact formulation of a cardinality-constrained graph partitioning problem with supervision, *a non-degenerate* problem in the unlabeled data limit. We also prove sufficient conditions for exact recovery of integer solutions.

2. Numerically, we develop a scalable and convergent iterative method based on ADMM and demonstrate superior performance and scalability in low, medium, high, and imbalanced label rate and data regimes on $k$-NN graphs (MNIST, Fashion-MNIST, and Cifar-10) as well as Planetoid citation networks and large-scale (160k-2.5M nodes) OGB graphs compared to state-of-the-art graph-based SSL algorithms.

3. We describe a connection to Laplace learning, and derive a simple and intuitive explanation for the efficacy of the mean-shift heuristic.

## 2 Preliminaries

In this section, we review graph semi-supervised learning, Laplace learning, the combinatorial minimum cut problem, and an associated continuous extension.

### 2.1 Laplace learning

Let $V = \{v_1, v_2, \ldots, v_M\}$ denote the $M$ vertices of the graph $G$ with weight matrix $W$ whose entries $w_{ij} \geq 0$ are the edge weights between $v_i$ and $v_j$. We assume the graph is symmetric, i.e., $w_{ij} = w_{ji}$. We define the degree $d_i = \sum_{j=1}^{n} w_{ij}$. Without loss of generality, we assume the first $m$ vertices $l = \{v_1, v_2, \ldots, v_m\}$ are assigned labels $\{\mathbf{y}_1, \mathbf{y}_2, \ldots, \mathbf{y}_m\}$, where $0 < m \ll M$. In the context of $k$-class classification we take each $\mathbf{y}_i$ to be one of the $k$ standard basis vectors $\{\mathbf{e}_1, \mathbf{e}_2, \ldots, \mathbf{e}_k\}$ of the form $\mathbf{e}_i = (0, \ldots 0, 1, 0, \ldots, 0)$, i.e. a one-hot row vector. Let $n$ denote the number of unlabeled vertices, i.e. $n = M - m$. The problem of graph-based semi-supervised learning is to smoothly propagate the labels over the unlabeled vertices $u = \{v_{m+1}, v_{m+2}, \ldots, v_M\}$.

Given a graph and a set of labeled vertices, the solution of Laplace learning [38], is the minimizer of the following quadratic program with label constraints $(X_0)_i = y_i$:

$$\min_{X_0 \in \mathbb{R}^{M \times k}} \left\{ \text{tr}(X_0^\top \mathcal{L} X_0) : (X_0)_i = y_i, \ 1 \le i \le m \right\}, \tag{1}$$

where $X_0 \in \mathbb{R}^{M \times k}$, $\mathcal{L}$ is the combinatorial graph Laplacian given by $\mathcal{L} = D - W$, where $D = \text{diag}(W \mathbf{1}_M)$ is a diagonal matrix whose elements are the node degrees, The prediction for vertex $v_i$ is determined by a heuristic of thresholding the largest component of $(X_0)_i$:

$$\arg\max_{j \in \{1, \dots, k\}} \{(X_0)_{ij}\}. \tag{2}$$

Note that Laplace learning is also called *label propagation (LP)* [39], since the Laplace equation associated with (1), can be solved by repeatedly replacing $(X_0)_i$ with a weighted average of its neighbors.

Let $\mathcal{L} = \left[ \begin{smallmatrix} L_l & L_{lu} \\ L_{ul} & L_u \end{smallmatrix} \right]$ and $X_0 = \left[ \begin{smallmatrix} X_l \\ X_u \end{smallmatrix} \right]$ where $l$ and $u$ correspond to labeled and unlabeled indices, respectively. For brevity, we denote $X = X_u$ and $Y = X_l$. The solution $\hat{X}$ satisfies the linear system

$$L_u \hat{X} = B := -L_{ul} Y. \tag{3}$$

Although Laplace learning and related algorithms work well when the number of labeled examples is sufficient, these algorithms generally suffer from two drawbacks: (1.) they typically solve a relaxation such that rather than predicting categorical values (e.g., binary one hot-encodings), they assign a real value for each class. A heuristic (usually thresholding) is necessarily applied to determine the predicted categorical label. (2.) In the low label rate regime, Laplace learning degenerates, yielding homogeneous predictions. Thus, thresholding results in poor prediction. In contrast, we propose an approach derived from a *non*-degenerate problem and coupled with a method to ensure exact combinatorial predictions, removing the need for heuristic thresholding.

## 2.2 Graph partitioning

Given a graph $G = (W, V)$, consider the discrete, cardinality-constrained graph partitioning problem

$$\min_{P \subseteq V} \left\{ \text{Cut}(P, P^c) := \sum_{v_i \in P} \sum_{v_j \in P^c} w_{ij} \right\} \ \text{s.t.} \ |P| = m_1 \tag{4}$$

Critically, we note that this combinatorial problem is non-degenerate when $m \ll n$, if $m_1 < n$, due to the cardinality constraint on the solution [30]. Algorithms and solutions to this problem have been thoroughly studied, particularly in the context of neighborhood graphs and for formulations that replace the constraint $|P| = m_1$ with a penalty on the imbalance between partitions.

The applications of graph partitioning-inspired methods to data science are well-known, particularly various relaxations of graph cut problems, including spectral methods [31, 4], combinatorial algorithms [21, 13], and semi-definite programs (SDPs) [16, 25]. However, key drawbacks of these methods include the inexactness of the relaxation (spectral methods), high computational price due to a large number of constraints (SDPs), or slow convergence (combinatorial algorithms).

On the other hand, while exact continuous nonconvex formulations for graph partitioning have been studied [17], as far as we are aware, no attempts have been made to design *efficient* algorithms amenable to $k$-way partitioning of large-scale datasets *with partial label information*. In particular, the $k$-way graph cut generalization was studied in [17]. The developed algorithms are based on a piecewise linear refinement of an initial, continuous-valued solution to one that is integer-valued and a local minimizer of a certain nonconvex relaxation (Theorem 2.1 in their paper). These algorithms guarantee an integer solution that exhibits a valid cut with objective value greater than that of the continuous-valued extension used to initialize their method. However, note that these algorithms are prohibitively expensive, e.g., potentially requiring exhaustive search over $3^{n^2}$ feasible "path matrices".

## 2.3 Graph cuts and continuous quadratic programming

Here we restate the discrete graph partitioning problem in a form more amenable for continuous quadratic optimization, i.e. as a continuous optimization problem over $\mathbf{x} \in \mathbb{R}^M$. Later, we will see

that this problem bears similarities to the Laplace learning problem in that the two share the same objective, but the graph partitioning problem involves additional constraints.

For simplicity, we first introduce a bipartitioning framework, $V_0(\mathbf{x}) := \{v_i : \mathbf{x}_i = 0\}$, $V_1(\mathbf{x}) := \{v_j : \mathbf{x}_j = 1\}$ characterized by a binary vector $\mathbf{x} \in \{0, 1\}^M$. Equivalently, each binary vector $\mathbf{x}$ determines an edge set $E_x = \{(i, j) : \mathbf{x}_i = 0, \mathbf{x}_j = 1\}$ connecting two subsets $V_0(\mathbf{x})$, $V_1(\mathbf{x})$. The cardinality-constrained min-cut solution is the binary vector $\mathbf{x}$ satisfying $\sum_i \mathbf{x}_i = m_1$

$$\underset{\mathbf{x} \in \{0,1\}^M}{\arg\min} \left\{ (\mathbf{1}_M - \mathbf{x})^\top W \mathbf{x} \right\} = \underset{\mathbf{x} \in \{0,1\}^M}{\arg\min} \sum_{(i,j) \in E_x} w_{ij}. \tag{5}$$

The equality is due to both terms being equal, i.e. $(\mathbf{1}_M - \mathbf{x})^\top W \mathbf{x} = \sum_{(i,j) \in E_x} w_{ij}$, over the set of binary vectors. Next, note that when $\mathbf{x}$ is binary, $(\mathbf{1}_M - \mathbf{x})^\top S \mathbf{x} = 0$ for any diagonal matrix $S$. Hence, min-cut is equivalent to

$$\underset{\mathbf{x} \in \{0,1\}^M}{\min} \left\{ (\mathbf{1}_M - \mathbf{x})^\top (W + S) \mathbf{x} \right\} \tag{6}$$

The theoretical aspects of this perturbed problem were investigated in [17]. However, an algorithm to solve this problem was left as future work.

For completeness, we review the multi-partition generalization before proposing an efficient iterative scheme in Section 3. In particular, the choice of $S$ is characterized in [17] as a way to provide a tighter relaxation from the set of binary vectors to the real-valued box set $[0, 1]^M$. Essentially, the term $x^\top S x$ acts as a regularizer. For instance, take $S = I$. The maximizer of the quadratic $x^\top S x = x^\top x = ||x||_2^2$ on the simplex $\{x \in [0, 1] : \sum_i x_i = 1\}$ are the extreme points of the simplex, i.e. the one-hot vectors. Hager and Krylyuk [17] rigorously proved that the larger the entries of $S$, the tighter the relaxation, but the number of stationary points grows. The key condition that must be satisfied by $S$ is stated in Theorem 2.1 of [17]: $S_{ii} + S_{jj} \geq 2w_{ij}$ which ensures that the min-cut problem with cardinality constraint on $V_1$ coincides with the following *continuous* quadratic optimization problem,

$$\underset{\mathbf{x} \in \mathbb{R}^M}{\min} (\mathbf{1}_M - \mathbf{x})^\top (W + S) \mathbf{x} \quad \text{s.t. } 0 \leq \mathbf{x}_i \leq 1, \quad \mathbf{1}_M^\top \mathbf{x} = m_1 = |V_1| \tag{7}$$

Given $S \neq 0_{n \times n}$, an essential question is how to compute a solution to (7). Note that it is no longer a convex problem. Intuitively, given some quadratic (e.g., parameterized by $W$) with a minimum in the interior of the simplex, perturbing $W$ by $S$ results in a new, indefinite or concave quadratic with a minimum at a vertex of the simplex.

Our proposed framework is based on constructing a homotopy path, e.g. with $S = sD$ for some parameter $s$ in (7), to find high-quality stationary points at extreme points of the simplex. In other words, our proposed method computes $x(s_0)$ for some small $s_0$, and then uses $x(s_0)$ as initialization to reach $x(s)$ for a larger value $s$, until solutions are within some small radius $\epsilon$ of an integral solution. By controlling the magnitude of $s$, we control the number of critical points that trap iterative methods.

We illustrate these principles in Figure 1, where we visualize the level sets of the 2-d quadratic $f(x, y) = (x - 0.8)^2 - (y - 0.2)^2 - s(x^2 + y^2)$ on the simplex $\{(x, y) \in [0, 1]^2 : x + y = 1\}$ for various choices of $s$. Note that when $s = 0$, the quadratic is convex and the minimum lies in the interior of the simplex. As $s$ increases, the quadratic becomes indefinite, and eventually concave on the simplex (i.e. when $s = 10$). As $s$ increases, the local minima on the simplex gravitate towards the "corners" (one-hot labels) until both "corners" become local minima in the extreme case.

## 3 Graph Partitioning for SSL

In this section, we generalize semi-supervised bi-partitioning (2-class SSL) to $k$-way partitioning (multi-class SSL). This yields a constrained optimization problem over $n \times k$ real-valued matrices. We generalize the previous framework to incorporate label information, introduce the conditions necessary for recovering binary-valued solutions, and develop our CutSSL algorithm.

### 3.1  Semi-supervised $k$-way partitioning

Before introducing label information, we transition from the bipartitioning to the $k$-way partitioning setting. The following maximization problem, introduced in [17], is interpreted as the maximization

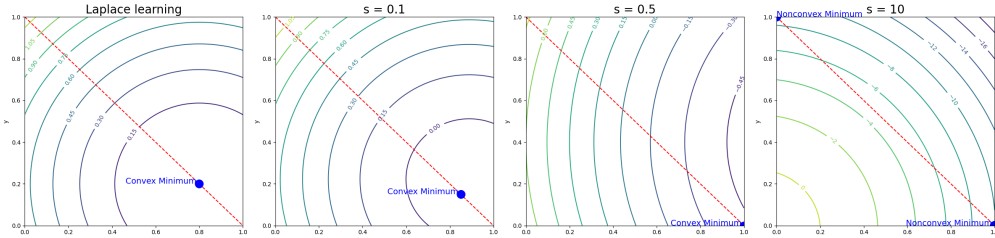

Figure 1: Effect of perturbing $L$ by $S$ on minimizers of (10) on the simplex. Shade of level-curves denotes descent direction. Blue dots denote unique minimizers in the simplex. As the magnitude of the perturbation increases, the minimzers gravitate towards the extreme points of the simplex.

of the sum of intra-partition edge weights.

$$\max_{X \in \mathbb{R}^{M \times k}} \mathrm{tr}(X^\top (W + S)X) \quad \text{s.t. } X^\top \mathbf{1}_M = \mathbf{m}, \ X\mathbf{1}_k = \mathbf{1}_M, \ X \geq 0. \tag{8}$$

Here, the entries of the vector $\mathbf{m} \in \mathbb{R}^k$ denote the given cardinalities of *each* class and $\mathbf{1}_k^\top \mathbf{m} = M$, the number of all examples.

To motivate the introduction of labels to the cardinality-constrained cut problem, we relate the objective of (8) with the *perturbed* Laplacian quadratic form:

**Proposition 3.1** (Equivalent binary minimizers). *Let $H_S(X) = tr(X^\top (\mathcal{L} - S)X)$ be the quadratic form associated with the Laplacian, and let $G_S$ be the objective of (8), $G_S = tr(X^\top (W + S)X)$. Then, we have that*

$$\arg\max\{G_S(X) : X \in \Omega\} = \arg\min\{H_S(X) : X \in \Omega\}$$

*for all $X_\Omega \in \Omega$, where $\Omega = \{X \in \mathbb{R}^{M \times k} : X_{i,j} \in \{0,1\}, \quad X^\top \mathbf{1}_M = \mathbf{m}\}$.*

*Proof.* Observe that
$$H_S(X) + G_S(X) = \langle X, DX \rangle = ||D\mathbf{1}_M||_1 \tag{9}$$
is a constant and the minimizers and maximizers coincide. $\square$

The Laplace learning solution (3) alongside this proposition motivates the following Laplacian quadratic objective with perturbation $S$, which we term CutSSL. This is our key formulation.

$$\min_{X \in \mathbb{R}^{n \times k}} \left\{ F_s(X) := \frac{1}{2}\mathrm{tr}(X^\top (L_u - S)X) - \mathrm{tr}(X^\top B) \right\} \tag{10}$$
$$\text{s.t.} \quad X^\top \mathbf{1}_n = \mathbf{m}, \ X\mathbf{1}_k = \mathbf{1}_n, \ X \geq 0,$$

In practice for the homotopy path, we take $S = sD$, where we have for any feasible $X_\Omega \in \Omega$,

$$F_s(X_\Omega) = F_0(X_\Omega) - s||D^{1/2}X_\Omega||_F^2 = F_0(X_\Omega) - s||Dm||_1$$

Thus, any binary minimizer of (10) with $s = 0$ is also a binary minimizer with $s > 0$. For brevity denote $L_{u,s} = L_u - sD$. For real-valued $X$, we have that the objective is $F_s(X) = F_0(X) - s||D^{1/2}X||_F^2$. The quadratic term of the objective $F_s$ satisfies

$$\mathrm{tr}(X^\top L_{u,s}X) = \mathrm{tr}(X^\top ((1-s)D_u - W_u)X) = (1-s)\sum_i (d_u)_i ||X_i||_2^2 - \sum_{ij}(w_u)_{ij}\langle X_i, X_j\rangle.$$

Thus, we have the following natural interpretation:

- When $s = 0$, the objective mirrors Laplace learning.

- When $0 \leq s \leq 1$, the constrained minimizer of the quadratic encourages solutions with predictions at unlabeled high-degree vertices to have small norm.

- When $s > 1$, $L_{u,s}$ could be indefinite or negative definite. The minimizers are extreme points of the feasible set and correspond exactly to labels.

When $L_{u,s}$ satisfies a certain condition, recovery of exact binary solutions is guaranteed, i.e. the relaxation is exact if $(s-1)(D_{ii} + D_{jj}) \geq 2w_{ij}$. The following proposition characterizes the binary solutions to (10) with respect to an appropriate choice of $s$. The proof is provided in the Appendix.

**Proposition 3.2.** *Suppose $s$ is chosen such that $(L_{u,s})_{ii} + (L_{u,s})_{jj} < 2(L_{u,s})_{ij}$ for all pairs $(i, j)$. Then every local minimizer of $F_s(X)$ is a $0 - 1$ solution.*

Note that the condition is weaker than a negative definite condition of $L_{u,s}$. For instance, as $s \to \infty$ every extreme point of the feasible set of (10) (every feasible $0 - 1$ assignment) becomes a local solution. With smaller $s$, fewer of these extreme points are local minimizers.

### 3.2 CutSSL algorithm

The problem introduced in (10) is amenable to the alternating direction method of multipliers (ADMM) framework [8], a flexible approach to solve non-smooth and nonconvex optimization problems by introducing "coupled" subproblems derived from a "splitting" of the primal variables. We consider the following "split" problem, introducing the variable $T \in \mathbb{R}^{n \times k}$:

$$\min_{X, T \in \mathbb{R}^{n \times k}} \left\{ F(X, T) = \frac{1}{2}\text{tr}(X^\top (L_{u,s})X) - \text{tr}(X^\top B) \right\}$$
$$\text{s.t. } X^\top \mathbf{1}_n = \mathbf{m}, \ X\mathbf{1}_k = \mathbf{1}_n, \ T \geq 0, \ X = T \tag{11}$$

Define the Lagrangian, $G(X, T, \Lambda)$ to be the objective of the following problem:

$$\max_{\Lambda \in \mathbb{R}^{n \times k}} \min_{X, T \in \mathbb{R}^{n \times k}} \frac{1}{2}\text{tr}(X^\top L_{u,s}X) - \text{tr}(X^\top B) + \text{tr}(\Lambda^\top (X - T)) + \frac{1}{2}||X - T||^2$$
$$\text{s.t. } X^\top \mathbf{1}_n = \mathbf{m}, \ X\mathbf{1}_k = \mathbf{1}_n, \ T \geq 0. \tag{12}$$

In this instance, ADMM is implemented by searching for saddle points of $G$ via the following iterates:

$$X_{t+1} = \arg\min_{X \in \mathbb{R}^{n \times k}} G(X, T_t, \Lambda_t) \quad \text{s.t. } X^\top \mathbf{1}_n = \mathbf{m}, \ X\mathbf{1}_k = \mathbf{1}_n$$
$$T_{t+1} = \arg\min_{T \in \mathbb{R}^{n \times k}} G(X_{t+1}, T, \Lambda_t) \text{ s.t. } T \geq 0 \tag{13}$$
$$\Lambda_{t+1} = \Lambda_t + X_{t+1} - T_{t+1}$$

The first-order optimality conditions on $X$ are

$$X = \bar{L}_{u,s}^{-1}(B + T - \Lambda - \mathbf{1}_n\mu_1^\top - \mu_2\mathbf{1}_k^\top), \tag{14}$$

where $\mu_1 \in \mathbb{R}^k$ and $\mu_2 \in \mathbb{R}^n$ are multipliers associated with the constraints $X^\top \mathbf{1}_n = \mathbf{m}$ and $X\mathbf{1}_k = \mathbf{1}_n$ respectively, and $\bar{L}_{u,s} = L_{u,s} + I$. The optimality conditions associated with $(\mu_1, \mu_2)$ are recovered by applying the constraint $X^\top \mathbf{1}_n = \mathbf{m}$. Let $c = \mathbf{1}_n^\top \bar{L}_{u,s}^{-1}\mathbf{1}_n$. This yields

$$\mu_1 = c^{-1}\{(B + T - \Lambda - \mu_2\mathbf{1}_k^\top)^\top \bar{L}_{u,s}^{-1}\mathbf{1}_n - \mathbf{m}\}$$
$$\mu_2 = \frac{1}{k}\{(B + T - \Lambda - \mathbf{1}_n\mu_1^\top)\mathbf{1}_k - \bar{L}_{u,s}\mathbf{1}_n\} \tag{15}$$

The optimality condition for $T$ yields

$$T = \arg\min_{T \geq 0} \frac{1}{2}||\Lambda + X - T||^2 \implies T = \max(\Lambda_t + X_{t+1}, 0) \tag{16}$$

The algorithm is summarized in Algorithm 1. Detailed derivations are provided in Appendix E.1.

---

**Algorithm 1** CutSSL

---

**Input:** Laplacian $L$, labels $B$, initialization $X_0$
**Output:** One-hot label predictions $X$

1: **function** ADMM($\bar{L}_{u,s}, X_0$)
2: $\quad X \leftarrow X_0$
3: $\quad$ **while** not converged **do**
4: $\quad\quad X_{t+1} = \bar{L}_{u,s}^{-1}\{B + T_t - \mathbf{1}_n\mu_1^\top - \mu_2\mathbf{1}_k^\top - \Lambda_t\}$ $\qquad\qquad$ ▷ $\mu_1, \mu_2$ given by (15)
5: $\quad\quad T_{t+1} \leftarrow \max(\Lambda_t + X_{t+1}, 0)$ $\qquad\qquad\qquad\qquad\qquad$ ▷ Projection of $T$
6: $\quad\quad \Lambda_{t+1} = \Lambda_t + X_{t+1} - T_{t+1}$ $\qquad\qquad\qquad\qquad\qquad$ ▷ update multipliers
7: $\quad$ **end while**
8: $\quad$ **return** $X_t$
9: **end function**

---

### 3.3 Convergence and complexity of CutSSL

The convergence of the standard two-block ADMM for convex and nonconvex problems has been thoroughly established in the literature [6, 8]. Note that when $s = 0$, (10) is convex and the feasible set is non-empty. An optimal solution exists. The ADMM iterations for (11) produce a sequence $\{(X_t, T_t, \Lambda_t)\}$, where $X_t$ is guaranteed to satisfy the equality constraints, the entries of $T_t$ are guaranteed to be positive and $\Lambda_t$ is the multiplier for equality constraint.

Generally, and as we show in the Appendix, ADMM converges to a KKT point of (11) if and only if the objective, primal, and dual residuals converge [8] as $t \to \infty$, i.e.

$$r_p^{t+1} = ||X_t - T_t||_2, \quad r_d^{t+1} = ||T_{t+1} - T_t||_2 \tag{17}$$

The iterations stop when the norms of these residuals are within specified tolerance levels. $\epsilon_p \geq 0$, $\epsilon_d \geq 0$. One may speed up the empirical rate of convergence of ADMM by adjusting a step-size parameter associated with the Lagrangian. See the Appendix for details.

Partitioning poses a challenge from an optimization perspective, primarily due to the nonconvexity of the quadratic objective. For example, the Poisson MBO method [10] is locally convergent in the limit and incurs complexity $O(TR)$, where $T$ is a bound on the number of iterations the procedure is run and $O(R)$ is the cost of solving a Laplacian system. Likewise, the volume MBO method presented in [19] also claims local convergence and complexity $O(TNV(\log(V) + N)C)$ for some constant $C$. Preconditioned conjugate gradient can be applied to find solutions to Laplacian-like systems in nearly linear time (linear in $M$) [29]. Thus, the complexity of our method is dominated by the primal variable update $O(TR)$ (step 4 in Alg. 1).

## 4 Mean-shifted Laplace learning

In the previous section, we introduced a graph-cut-inspired framework for graph-based semi-supervised learning. Here we consider a related problem formulation to (10) such that we do not limit our solutions to lie on the simplex, and set $s = 0$. This is equivalent to imposing a cardinality constraint on Laplace learning (1):

$$\min_{X \in \mathbb{R}^{n \times k}} \left\{ \frac{1}{2}\text{tr}(X^\top L_u X) - \text{tr}(X^\top B) \right\} \text{ s.t. } X^\top \mathbf{1}_n = \mathbf{m} \tag{18}$$

This comparison is justified with empirical evidence on MNIST. At solutions to (10) with $s = 0$, we observe (1.) the inequality constraints $X \geq 0$ are *inactive* (2.) the multiplier $\mu_2$ associated with the constraint $X\mathbf{1}_k = \mathbf{1}_n$ has infinitesimal norm (empirically order of $10^{-9}$) (3.) the solutions to (18) and (10) differ in norm by a tiny amount (order of $10^{-4}$).

We will show that an approximate solution to this problem is a previously known heuristic, mean-shift Laplace learning. Our analysis of this problem provides new evidence to explain the empirical performance of this heuristic, while also revealing that it is suboptimal in some sense.

While Laplace learning (3) exhibits excellent results at medium and higher label rates, recent work [10, 11] has identified that the solution to the Laplace learning problem degenerates to a constant as $m \ll M$, i.e., as the number of unlabeled vertices is significantly larger than the number

of labeled vertices. Specifically, in the low label rate regime, $\hat{X}$ asymptotically depends only on the degrees of the labeled vertices of the graph [10, 32]: $\hat{X}_i \approx \bar{y}_w := \frac{\sum_{i=1}^m d_i y_i}{\sum_{j=1}^m d_j}$, thus, thresholding as in (2) results in a poor solution, concentrated on a single class.

In the nonasymptotic regime, however, $\hat{X}$ is not exactly a constant, and earlier work has demonstrated that shifting $\hat{X}$ such that the column-means are zero (i.e. the *mean-shift*) empirically corrects this issue. This is justified in Calder et al. [10], von Luxburg et al. [32] via a random walk argument.

## 4.1 Mean-shift as an approximate solution for cardinality-constrained SSL

The problem in (18) is convex since the matrix $L_u$ is a principal submatrix of $\mathcal{L}$, a positive semidefinite matrix. Additionally, the set $C_{\mathbf{m}} = \{X : X^\top \mathbf{1}_n = \mathbf{m}\}$ is convex and nonempty. Mean-shifted Laplace learning is one heuristic to solve this problem, by performing the the following pair of steps:

1) **Linearization**: Solve $L_u X = B$ to get $\hat{X}$. This is equivalent to vanilla Laplace learning.
2) **Projection**: Project $\hat{X}$ onto the set $C_m$:

$$\hat{\hat{X}} = \arg\min_X \left\{ \frac{1}{2} \|X - \hat{X}\|_F^2 \text{ s.t. } X^\top \mathbf{1}_n = \mathbf{m} \right\},\tag{19}$$

Applying Lagrange multipliers yields $\hat{\hat{X}}_{ij} = \hat{X}_{ij} - \frac{1}{n}\left( \sum_{i=1}^n \hat{X}_{ij} - \mathbf{m}_j \right)$. Thus, mean-shift is the projection $\hat{\hat{X}}_{ij}$ when $\mathbf{m} = \mathbf{0}_k$.

This is only an approximate solution. However, it is straightforward to characterize the optimal solutions of (18), which can be decomposed into the sum of two terms: $X = Z + \frac{1}{n}\mathbf{1}_n\mathbf{m}^\top$. The first term is associated with the solution to (18) for $\mathbf{m} = 0$, i.e. $\mathbf{1}_n^\top Z = 0$. Denote the projection $P = I - \frac{1}{n}\mathbf{1}_n\mathbf{1}_n^\top$ onto the set of $n \times k$ matrices with mean-zero columns. Note that $Z$ is the minimizer of $\frac{1}{2}\text{tr}(Z^\top P L_u P Z) - \text{tr}(Z^\top(PB))$ and satisfies the system $Z = (PL_uP)^\dagger PB$. The second term $\frac{1}{n}\mathbf{1}_n\mathbf{m}^\top$ is a constant term to correct for the true value of $\mathbf{m}$ so that $\mathbf{1}_n^\top X = \mathbf{m}$. Thus, it is clear that to resolve the suboptimality of the mean-shift heuristic, one must apply mean-shift to the Laplace learning solution when the *labels have been shifted*. This can be solved with Projected Conjugate Gradient (PCG). In the following proposition, we characterize optimal solutions to (18) in terms of the mean-shift heuristic. Naturally, we expect that as the number of labeled vertices increases, the gap becomes smaller. The proof is provided in the appendix.

**Proposition 4.1.** *Let $X^* \in \mathbb{R}^{n \times k}$ be a solution to Laplace learning (18), $\hat{X} \in \mathbb{R}^{n \times k}$ be the solution to (3), and $\hat{\hat{X}} \in \mathbb{R}^{n \times k}$ be the mean-shift heuristic (28). Let $\kappa(L_u) = \frac{\lambda_{max}(L_u)}{\lambda_{min}(L_u)}$ and $\hat{\mathbf{m}} = \mathbf{1}_n\hat{X}$. Then, $X^*$ is a rank-one perturbation of $\hat{\hat{X}}$ and $\|X^* - \hat{\hat{X}}\| \leq \frac{\kappa(L_u)}{\sqrt{n}}\|\mathbf{m}^\top - \hat{\mathbf{m}}^\top\| + \frac{1}{n}\|\hat{\mathbf{m}}\|$.*

In the projection step, the rank-one perturbation (33) serves to adjust the column sum of $\hat{X}$ to satisfy the cardinality constraint implied by $\mathbf{m}$. When thresholding continuous solutions to integer predictions by comparing the entries in each row of $X$, the rank-one adjustment can play a crucial balancing role, particularly when $\hat{\mathbf{m}}$, the cardinalities of the predicted labels, are far from the prior cardinalities $\mathbf{m}$. We emphasize that the term $\hat{\mathbf{m}}$ exactly corresponds to the degeneracy of Laplace learning $\hat{X}$.

Notably, in the low-label rate regime, the bound is dominated by the smallest eigenvalue of $L_u$, the denominator of $\kappa(L_u)$. This term is intimately related to the number of labeled vertices, i.e. by interlacing [14], as well as the difference in cut and cut quality, by the above proposition. Generally, the bound gets smaller as the smallest eigenvalue of $L_u$ increases, and this corresponds to the addition of labeled samples to the label set.

In the appendix, we empirically confirm this in Fig 3, where we investigate solutions generated using the mean-shift Laplace learning heuristic and the true solution to (18) on MNIST at label rates ranging from 1 per-class to 10 per-class.

Despite not resolving the issue of degeneracy in the theoretical sense of Calder et al. [11], we show in Sec. 5 that solving (18) exactly via PCG consistently yields better results than the mean shift heuristic, at no additional computational cost. Intuitively, our analysis implies that shifting *both* the solution

*and* the labels is a superior heuristic. However, we note that while gains in accuracy of $1 - 2\%$ are consistent in the very low label-rate regime, the improvement becomes marginal at higher label rates. In contrast, we demonstrate that CutSSL, which avoids degenerate solutions via recovery of integer solutions, has consistent improvement in performance across label rates.

## 5 Experiments

We evaluate our method on three image datasets: MNIST [12], Fashion-MNIST [33] and Cifar-10 [23], using pretrained autoencoders as feature extractors as in Calder et al. [10], see appendix for details. We also evaluate cutSSL on various real-world networks, including the Cora citation network and the OGB [18] Arxiv and Product networks.

Table 1: Cifar-10. Average accuracy over 100 trials with standard deviation in brackets.

| CIFAR-10 # LABELS PER CLASS | 1 | 3 | 5 | 10 | 4000 |
|---|---|---|---|---|---|
| LAPLACE/LP [38] | 10.4 (1.3) | 11.6 (2.7) | 14.1 (5.0) | 21.8 (7.4) | 80.9 (0.0) |
| MEAN SHIFT LAPLACE/LP | 40.9 (4.1) | 49.5 (3.4) | 51.3 (2.9) | 57.0 (2.1) | 81.0 (0.4) |
| EXACT LAPLACE/LP (OURS) | 41.6 (4.3) | 50.1 (2.9) | 51.6 (2.6) | 57.3 (1.9) | 81.1 (0.2) |
| LE-SSL [3] | 36.2 (0.1) | 50.2 (4.3) | 54.7 (3.4) | 59.4 (2.3) | 80.1 (0.9) |
| SPARSE LP [20] | 13.1 (2.9) | 13.4 (2.6) | 18.4 (2.1) | 19.8 (1.9) | 71.3 (0.1) |
| POISSON [10] | 40.7 (5.5) | 49.9 (3.4) | 53.8 (2.6) | 58.3 (1.7) | 80.3 (0.9) |
| VOLUME-MBO [19] | 38.0 (7.2) | 50.1 (5.7) | 55.3 (3.8) | 59.2 (3.2) | 75.1 (0.2) |
| POISSON-MBO [10] | 41.8 (6.5) | 53.5 (4.4) | 57.9 (3.2) | 61.8 (2.2) | 80.1 (0.3) |
| **CUTSSL (OURS)** | **44.7 (5.9)** | **56.1 (3.7)** | **59.7 (3.1)** | **64.5 (1.7)** | **82.0 (0.2)** |

**Image datasets** We construct a graph over the latent feature space. We used all available data to construct the graph, with $n = 70,000$ nodes for MNIST and Fashion-MNIST, and $n = 60,000$ nodes for Cifar-10. The graph was constructed as a $k$-nearest neighbor graph with Gaussian edge weights given by $w_{ij} = \exp\left(-4||v_i - v_j||^2/d_k(v_i)^2\right)$, where $v_i$ are the latent variables for image $i$, and $d_k(v_i)$ is the distance in the latent space between $v_i$ and its $k^{\text{th}}$ nearest neighbor. We used $k = 10$ in all experiments and symmetrize $W$ by replacing $W$ with $\frac{1}{2}(W + W^\top)$.

We compare to the state-of-the-art unconstrained graph-based SSL algorithm Poisson Learning [10] as well as variants based on the MBO procedure [19] to incorporate knowledge of class sizes. We include vanilla Laplace learning, Laplace learning with the mean-shift heuristic, and exact Laplace learning with relaxed cardinality constraints presented in Section 4. We also include a method inspired from spectral relaxations of the min-cut problem. Laplacian Eigenmaps-SSL [3] performs graph-based SSL by learning a linear predictor using the principal eigenvectors of the graph Laplacian as features. We note that our method is directly comparable to Volume- and Poisson-MBO, which also incorporate knowledge of class sizes.

Adopting a homotopy method implies that we solve a sequence of quadratic programs with $s_0 = 0$ and use the associated solution to initialize a subsequent problem with $s > 0$. In practice, we set $s = 0.1$ for all experiments. The procedure presented in Alg. 1 is run for 100 iterations. In Table 1 we present the accuracy of our method on Cifar-10 across various label rates (MNIST and fMNIST results are in the appendix). CutSSL outperforms all methods across all label rates. In particular, we outperform the state-of-the-art by $2.9\%$ when only a single sample from each class is provided and by $1.9\%$ in the large label-rate regime.

Table 2: A citation network with $169,343$ vertices and $40$ labels ("Rate=tr" means all train data).

| OGBN-ARXIV, $k = 40$ | ACC. (RATE=1) | RUNTIME (S) | CUT ($10^3$) | ACC. (RATE=TR) |
|---|---|---|---|---|
| POISSON [10] | 45.72% | **79.34** | 71.13 | 50.11% |
| GRAPHHOP-NN [34] | 27.36% | 1878.39 | 94.16 | 34.47% |
| POISSON-MBO [10] | 46.94% | 341.12 | 72.09 | 51.7% |
| **CUTSSL (OURS)** | **52.37%** | 122.04 | **70.43** | **68.21%** |

Table 3: A co-purchasing network with $>2M$ vertices and $47$ labels ("Rate=tr" means all train data).

| OGBN-PRODUCTS, $k = 47$ | ACC. (RATE=1) | RUNTIME (S) | CUT ($10^3$) | ACC. (RATE=TR) |
|---|---|---|---|---|
| POISSON [10] | 48.31% | 113.91 | 117.21 | 52.14% |
| GRAPHHOP-NN [34] | 19.47% | 3490.64 | 163.24 | 31.16% |
| P-LAPLACE [15] | 43.07% | **91.01** | 132.93 | 52.11% |
| POISSON-MBO [10] | 49.07% | 412.01 | 91.13 | 56.31% |
| **CUTSSL (OURS)** | **54.22%** | 188.24 | **94.21** | **61.24%** |

**Citation networks :** In Figure 2 we plot accuracy and label rate for the Cora and Citeseer citation networks, demonstrating that our method generalizes beyond $k$-NN graphs. Notably, for Cora, we outperform PoissonMBO by $7.5\%$ at 1 label per class (see appendix). We note that the trend persists across datasets and label rates. Interestingly, LE-SSL [3] performs significantly worse across label rates. This implies that searching near the set of binary vectors is critical to achieving good predictors.

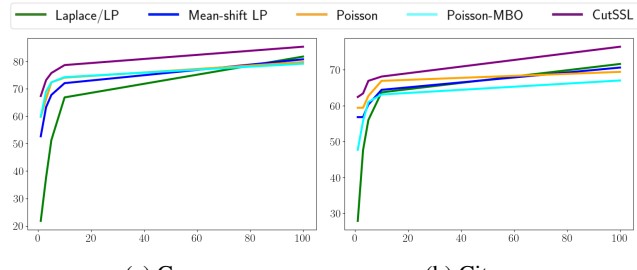

Figure 2: Prediction accuracy on citation networks.

**Large-scale networks:** To demonstrate the scalability of our method, we provide results at 1-label rate for OGB networks Arxiv (Table 2) and Products (Table 3). These networks contain up to 2 million vertices and $47$ classes. We demonstrate state-of-the-art performance in this regime, with an improvement of $\sim 5\%$ for both. Note that CutSSL also significantly outperforms a graph neural network-based method, GraphHop [34], designed for low label rates and which also exploits node feature information. CutSSL has double the accuracy with an order of magnitude less run-time.

**Label/class imbalance:** In the appendix, we also present results demonstrating performance under label and class imbalance. Our method exhibits superior robustness when either the label rate is imbalanced (different classes have different number of labels, see Tab. 5) or when the underlying partition sizes are imbalanced (different classes have different number of samples, see Tab. 6).

**Additional experiments:** In the appendix (section C.2), we demonstrate the efficacy of CutSSL at classifying samples with small "margin". This is in contrast to spectral and Laplace learning-type algorithms are known to produce predictions that "bleed" across the cut boundary. We conduct ablative studies on the graph construction and choice of $S$ (section C.4). Convergence of CutSSL and recovery of integer solutions is empirically demonstrated in Figure 6.

## 6    Conclusion

We have proposed a novel formulation of graph-based semi supervised learning as a nonconvex continuous quadratic program that bears similarities to the mean-shift Laplace learning heuristic and graph cuts. We have presented an iterative method to solve a sequence of these problems to recover *discrete* predictions. Numerically, we have demonstrated that our approach consistently outperforms state-of-the-art methods on semi-supervised learning problems at low, medium, and high label rates and in imbalanced class regimes. Future work includes a rigorous analysis of exact cut-based methods for graph-based semi-supervised learning. Of particular interest are the asymptotic behavior and consistency of cut-based methods for graph-based SSL.

## Acknowledgments and Disclosure of Funding

This work was partially funded by NSF award 2217058, an award from the W.M. Keck foundation, and NSTC grant 110-2115-M-005-007-MY3.

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

# A   Outline of appendix

The outline of the appendix is as follows: we first provide the proofs of the propositions presented in Sections 3 and 4. Next, we provide additional experiments to demonstrate the efficacy of our method on other standard image and citation network benchmarks (FMNIST and FashionMNIST and Planetoid) and in the presence of imbalanced labels. We also include an ablation study of $S$ and the graph construction, and an evaluation of the cut objective for different methods. Additionally, we review two existing methods for cardinality-constrained semi-supervised learning. We also provide detailed derivations and a variation of the algorithm presented in the main text which incorporates a step size.

# B   Proofs

Consider the CutSSL formulation:

$$\min_{X \in \mathbb{R}^{n \times k}} \left\{ F(X) = \frac{1}{2} \text{tr}(X^\top (L_u - S)X) - \text{tr}(X^\top B) \right\} \tag{20}$$
$$\text{subject to } X^\top \mathbf{1}_n = \mathbf{m}, \ X\mathbf{1}_k = \mathbf{1}_n, \ X \geq 0.$$

The following proposition characterizes the binary solutions to (20) assuming appropriate choice of $s$.

**Proposition B.1** (Proposition 3.2). *Suppose $s$ is chosen such that $(L_{u,s})_{ii} + (L_{u,s})_{jj} < 2(L_{u,s})_{ij}$ for all pairs $(i,j)$. Then every local minimizer of $F$ is a $0-1$ solution.*

*Proof.* Suppose $X$ is a non-binary optimal solution to (20). Assume that $X_{i_1, i_2}$ lies in $(0, 1)$. The feasibility condition implies that both $X_{i_3, i_2}$ and $X_{i_1, i_0}$ lie in the interval $(0, 1)$ for some $i_3$ and some $i_0$. Likewise, $X_{i_4, i_3} \in (0, 1)$ for some $i_4$. Repeat the argument, then there exists a sequence of $l$ nonzero entries in $X$, whose indices form a cycle, i.e., $(i_1, i_2, i_3 \ldots . i_l)$ and $i_l = i_0$ with $l$ even. That is, $X_{i_j, i_{j+1}} \in (0, 1)$, $X_{i_{j+2}, i_{j+1}} \in (0, 1)$ for $j = 1, 3, 5, \ldots$. The existence of a cycle in $X$ implies that we can construct $X + \epsilon E$ feasible, where $E$ is a $0/1$ matrix with $E_{i_j, i_{j+1}} = 1$ and $E_{i_{j+2}, i_{j+1}} = -1$. Clearly, we have that

$$E\mathbf{1}_k = 0 \text{ and } E^\top \mathbf{1}_n = 0. \tag{21}$$

From the local optimality condition $f(X + \epsilon E) \geq f(X)$, with $\epsilon \to 0^+$, the necessary conditions indicate two conditions,

$$\text{tr}(E^\top (L_{u,s}X - B)) = 0, \ \text{tr}(E^\top L_{u,s}E) \geq 0 \tag{22}$$

for each $E$ with cycle $(i_1, \ldots, i_l)$. Since $X$ is a locally optimal solution, then $L_{u,s}X - B = \mu_1 \mathbf{1}_k^\top + \mathbf{1}_n \mu_2^\top$ holds for some vectors $\mu_1, \mu_2$, which ensures the condition $\text{tr}(E^\top (L_{u,s}X - B)) = 0$. However, according to the assumption $(L_{u,s})_{ii} + (L_{u,s})_{jj} < 2(L_{u,s})_{ij}$, we have that

$$\text{tr}(E^\top L_{u,s}E) = \sum_{j=1}^{l} 2(L_{u,s})_{i_j, i_j} - 2(L_{u,s})_{i_j, i_{j+1}} < 0 \tag{23}$$

i.e., the conditions for local optimality are violated. $\qquad \square$

Let $\mathcal{L} = \begin{bmatrix} L_l & L_{lu} \\ L_{ul} & L_u \end{bmatrix}$ and $X_0 = \begin{bmatrix} X_l \\ X_u \end{bmatrix}$ where $l$ and $u$ correspond to labeled and unlabeled indices, respectively. The solution $\hat{X}$ of the Laplace learning problem satisfies the linear system

$$L_u \hat{X} = B := -L_{ul} Y. \tag{24}$$

Consider the cardinality constrained Laplace learning formulation

$$\min_{X \in \mathbb{R}^{n \times k}} \left\{ f(X) = \frac{1}{2} \mathrm{tr}(X^\top L_u X) - \mathrm{tr}(X^\top B) \right\} \tag{25}$$
$$\text{s.t. } X^\top \mathbf{1}_n = \mathbf{m}$$

In the main text, we discussed that mean-shift Laplace learning is a heuristic that recovers approximate solutions to the above convex problem for $\mathbf{m} = \mathbf{0}_k$. The mean-shift heuristic corresponds to the following two-step procedure:

1) **Linearization**: Solve $L_u X = B$ to get $\hat{X}$
2) **Projection**: Project $\hat{X}$ onto the orthogonal complement of $\mathbf{1}_n$ to get $\hat{\hat{X}} = P\hat{X}$, where $P = I - \frac{1}{n} \mathbf{1}_n \mathbf{1}_n^\top$ First, we provide a detailed derivation of step 2.

We have

$$\hat{\hat{X}} = \arg\min_X \left\{ \frac{1}{2} \|X - \hat{X}\|_F^2 \text{ s.t. } X^\top \mathbf{1}_n = \mathbf{m} \right\}, \tag{26}$$

Let $\mathbf{m} = 0$. Applying Lagrange multipliers to (19) in order to solve step 2 yields

$$\mathcal{G}(X, \mu) = \frac{1}{2} \|X - \hat{X}\|_F^2 + \mu^\top (X^\top \mathbf{1}_n - \mathbf{m})$$
$$= \left( \frac{1}{2} \|X - \hat{X}\|_F^2 + (X\mu)^\top \mathbf{1}_n \right) - \mu^\top \mathbf{m} \tag{27}$$

with $\mu \in \mathbb{R}^k$. The solution to (19) is given by finding the $\mu$ which satisfies the constraint. The partial derivatives of the Lagrangian are

$$\frac{\partial \mathcal{G}}{\partial X_{ij}} = (X_{ij} - \hat{X}_{ij}) + \mu_j = 0 \implies X_{ij} = \hat{X}_{ij} - \mu_j$$

Applying the constraint $\sum_{i=1}^n X_{ij} = \mathbf{m}_j$ yields $\mu_j = \frac{1}{n}(\sum_{i=1}^n \hat{X}_{1ij} - \mathbf{m}_j)$. The projection of $\hat{X}$ onto $C_{\mathbf{m}}$ is

$$\hat{\hat{X}}_{ij} = \hat{X}_{ij} - \frac{1}{n} \left( \sum_{i=1}^n \hat{X}_{ij} - \mathbf{m}_j \right) \tag{28}$$

Here, we characterize the connection between this heuristic and the cardinality-constrained problem.

**Proposition B.2** (Proposition 4.1). *Let $X^* \in \mathbb{R}^{n \times k}$ be a solution to Laplace learning (18), $\hat{X} \in \mathbb{R}^{n \times k}$ be the solution to (3), and $\hat{\hat{X}} \in \mathbb{R}^{n \times k}$ be the mean-shift heuristic (28). Let $\kappa(L_u) = \frac{\lambda_{max}(L_u)}{\lambda_{min}(L_u)}$ and $\hat{\mathbf{m}} = \hat{X}^\top \mathbf{1}_n$. Then, $X^*$ is a rank-one perturbation of $\hat{\hat{X}}$ and $\|X^* - \hat{\hat{X}}\| \leq \frac{\kappa(L_u)}{\sqrt{n}} \|\mathbf{m}^\top - \hat{\mathbf{m}}^\top\| + \|\hat{\mathbf{m}}\|$.*

Consider the first order condition of (25):

$$L_u X^* = B + \mathbf{1}_n \mu^\top \tag{29}$$

for some multiplier $\mu \in \mathbb{R}^k$ satisfying

$$\mathbf{m} = \mu (\mathbf{1}_n^\top L_u^{-1} \mathbf{1}_n) + (B^\top L_u^{-1} \mathbf{1}_n) \tag{30}$$

i.e., due to the constraint $X^\top \mathbf{1}_n = \mathbf{m}$,

$$\mathbf{m} = (L_u^{-1}(B + \mathbf{1}_n \mu^\top))^\top \mathbf{1}_n$$
$$= (L_u^{-1} B + L_u^{-1} \mathbf{1}_n \mu^\top)^\top \mathbf{1}_n \tag{31}$$
$$= \mu (\mathbf{1}_n^\top L_u^{-1} \mathbf{1}_n) + (B^\top L_u^{-1} \mathbf{1}_n)$$

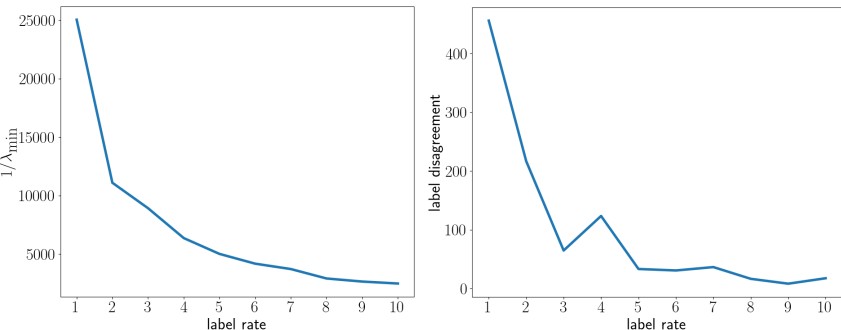

Figure 3: **(top)** Reciprocal of the small eigenvalue of $L_u$ **(bottom)** Label disagreements

The first order condition implies that $X^*$ is determined by

$$X^* = L_u^{-1} 1_n (1_n^\top L_u^{-1} 1_n)^{-1} \mathbf{m}^\top + \{I - L_u^{-1} 1_n (1_n^\top L_u^{-1} 1_n)^{-1} 1_n^\top\} L_u^{-1} B^\top \tag{32}$$

Let $\hat{X}$ denote the solution to the classic Laplace learning problem: $\hat{X} = L_u^{-1} B^\top$ and $\hat{\mathbf{m}} = 1_n^\top \hat{X}$. Also, let $y = L_u^{-1} 1_n$ and introduce the notation $1_n^\top L_u^{-1} 1_n = 1_n^\top y$. Then, we may show that $X^*$ is a rank-one perturbation of $\hat{X}$,

$$X = \hat{X} + (1_n^\top y)^{-1} y (\mathbf{m}^\top - \hat{\mathbf{m}}^\top) \tag{33}$$

$\square$

Informally, since $\mathcal{L}$ has a null vector $\mathbf{1}_n$ (smallest eigenvalue 0), in the low-label rate regime, $L_u$ is nearly singular and $L_u^{-1} \mathbf{1}_n$ corresponds a very large drift term.

*Remark* B.3. Denote the condition number of $L_u$ $\kappa(L_u) = \frac{\lambda_{\max}(L_u)}{\lambda_{\min}(L_u)}$. It follows that

$$\begin{aligned}
||(1_n^\top y)^{-1} y (\mathbf{m}^\top - \hat{\mathbf{m}}^\top)|| &\leq (1_n^\top y)^{-1} ||y|| \cdot ||\mathbf{m} - \hat{\mathbf{m}}|| \\
&\leq \frac{\kappa(L_u)}{\sqrt{n}} ||\mathbf{m} - \hat{\mathbf{m}}||
\end{aligned} \tag{34}$$

*Remark* B.4.

$$||X - (\hat{X} - \frac{1}{n} \hat{\mathbf{m}})|| \leq ||(1_n^\top y)^{-1} y (\mathbf{m}^\top - \hat{\mathbf{m}}^\top)|| + \frac{1}{n} ||\hat{\mathbf{m}}|| \leq \frac{\kappa(L_u)}{\sqrt{n}} ||\mathbf{m} - \hat{\mathbf{m}}|| + \frac{1}{n} ||\hat{\mathbf{m}}||$$

We note that $M$, the total number of vertices remains a constant. $M = n + m$, where $n$ are the number of unlabeled vertices and $m$ are the number of labeled vertices. The bound highlights the gap as $m$ varies, with $M$ fixed. As $m$ increases, $n$ decreases and $\kappa(L_u)$ increases (due to interlacing). Empirically, in Figure 3, we plot the difference in integer-rounded predictions and the reciprocal of the smallest eigenvalue of $L_u$. We note that the reciprocal of the smallest eigenvalue of $L_u$ decays with the label rate, as does the difference in predictions.

## C   Additional Details and Experiments

In this section, we provide additional numerical evidence for the efficacy of our method.

See the Colab link provided in the abstract for an implementation of the algorithm in Python and NumPy and the graphlearning package [9]. All experiments were evaluated on Colab instances. These instances were equipped with 2-core AMD EPYC 7B12 CPUs and 13 GB of ram.

We additionally explore failure cases of our method—examples of data which our method incorrectly classifies; a setting in which emphasizes an imbalance in the label rate for each class; and an ablation study on the choice of the diagonal perturbation matrix $S$.

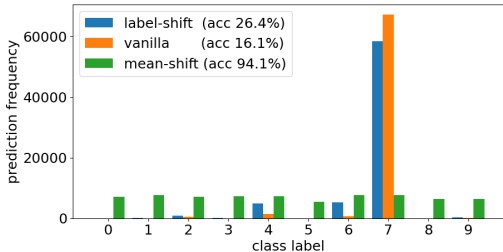

Figure 4: Predicted label distribution on MNIST at 1 label per class. Vanilla Laplace learning (orange) is degenerate. The mean-shift heuristic (green) imposes a constraint on the predictions corresponding to a balanced prior on the class distribution. Shifting the labels (blue) is insufficient to enforce the prior.

## C.1   MNIST and Fashion-MNIST

For MNIST and Fashion-MNIST, we used variational autoencoders with 3 fully connected layers of sizes (784,400,20) and (784,400,30), respectively, followed by a symmetrically defined decoder. The autoencoder was trained for 100 epochs on each dataset. The autoencoder architecture, loss, and training are similar to Kingma and Welling [22].

For each dataset, we construct a graph over the latent feature space. We used all available data to construct the graph, with $n = 70,000$ nodes for MNIST and Fashion-MNIST, and $n = 60,000$ nodes for Cifar-10. The graph was constructed as a $k$-nearest neighbor graph with Gaussian edge weights given by

$$w_{ij} = \exp\left(-4||x_i - x_j||^2/d_k(x_i)^2\right),$$

where $x_i$ are the latent variables for image $i$, and $d_k(x_i)$ is the distance in the latent space between $x_i$ and its $k^{\text{th}}$ nearest neighbor. We used $k = 10$ in all experiments and symmetrize $W$.

Table 4: Average accuracy over 100 trials with standard deviation in brackets. Best is bolded.

| MNIST # LABELS PER CLASS | 1 | 3 | 5 | 10 | 4000 |
|---|---|---|---|---|---|
| LAPLACE/LP [38] | 16.1 (6.2) | 42.0 (12.4) | 69.5 (12.2) | 94.8 (2.1) | 98.3 (0.0) |
| MEAN SHIFT LAPLACE/LP | 91.0 (4.7) | 95.7 (1.0) | 96.3 (0.5) | 96.7 (0.2) | 98.2 (0.1) |
| POISSON [10] | 90.2 (4.0) | 94.5 (1.1) | 95.3 (0.7) | 96.7 (0.2) | 97.2 (0.1) |
| VOLUME-MBO [19] | 89.9 (7.3) | 96.2 (1.2) | 96.7 (0.6) | 96.7 (0.2) | 96.9 (0.1) |
| POISSON-MBO [10] | 96.5 (2.6) | 97.2 (0.1) | 97.2 (0.1) | 97.6 (0.1) | 97.3 (0.0) |
| **CUTSSL** ($S = sI$) | 97.4 (0.1) | 97.6 (0.1) | 97.6 (0.1) | 97.6 (0.1) | 98.1 (0.1) |
| **CUTSSL** ($S = sD$) | **97.5 (0.1)** | **97.6 (0.1)** | **97.7 (0.1)** | **97.7 (0.1)** | **98.3 (0.1)** |
| FASHIONMNIST | | | | | |
| LAPLACE/LP [38] | 18.4 (7.3) | 44.0 (8.6) | 57.9 (6.7) | 70.6 (3.1) | 85.8 (0.1) |
| MEAN SHIFT LAPLACE/LP | 58.1 (5.1) | 67.6 (2.8) | 70.5 (2.1) | 74.4 (1.4) | 85.9 (0.2) |
| POISSON [10] | 60.8 (4.6) | 69.6 (2.6) | 72.4 (2.3) | 75.2 (1.5) | 82.4 (0.1) |
| VOLUME-MBO [19] | 54.7 (5.2) | 66.1 (3.3) | 70.1 (2.8) | 74.4 (1.5) | 75.1 (0.2) |
| POISSON-MBO [10] | 62.0 (5.7) | 70.4 (2.9) | 73.1 (2.7) | 76.1 (1.4) | 80.1 (0.3) |
| **CUTSSL** ($S = sI$) | 63.3 (4.8) | 70.4 (2.1) | 74.1 (1.2) | 76.2 (1.1) | 86.0 (0.1) |
| **CUTSSL** ($S = sD$) | **64.3 (4.6)** | **71.9 (3.8)** | **74.5 (1.3)** | **76.6 (1.0)** | **86.1 (0.1)** |

In Table 4 we present results at a variety of label rates on the MNIST and FashionMNIST datasets, demonstrating improved classification accuracy and lower classification variance accross all datasets.

To demonstrate the degeneracy of Laplace learning and the mean shift heuristic in the low label regime, we visualize the distribution of class predictions for MNIST given 1 labeled sampled per class in Figure 4. Vanilla Laplace learning predictions (orange) are concentrated on label '7', with very low prediction accuracy. Solving Laplace learning for shifted labels (blue) still shows an imbalanced class distribution, with a slight increase in performance, whereas mean-shifted Laplace learning (green) exhibits balanced class predictions and a significant increase in performance.

We demonstrate the robustness of CutSSL to two versions of "imbalance": imbalance in the label distribution (Tab. 5) and imbalance in the underlying class distribution (Tab. 6). Notably, our method exhibits superior robustness in both cases.

Table 5: Imbalanced label regime. Odd classes have 1 label, even classes have 5 labels.

| IMBALANCED LABELS | MNIST | FMNIST | CIFAR-10 |
|---|---|---|---|
| LAPLACE/LP | 69.5 | 21.1 | 10.0 |
| MEAN SHIFT LAPLACE | 91.3 | 62.6 | 37.2 |
| POISSON | 91.1 | 65.1 | 44.6 |
| POISSON-MBO | 92.7 | 66.2 | 55.0 |
| **CUTSSL (OURS)** | **93.2** | **68.7** | **59.4** |

Table 6: Imbalanced class regime. There are 10 times as many samples for each even class than there are for the odd classes, and one labeled node for each odd class.

| CLASS IMBALANCE | MNIST | FMNIST | CIFAR-10 |
|---|---|---|---|
| LAPLACE/LP | 54.7 | 38.6 | 18.4 |
| MEAN SHIFT LP | 92.5 | 64.3 | 42.8 |
| POISSON | 90.0 | 66.8 | 43.7 |
| POISSON-MBO | 91.8 | 77.5 | 52.3 |
| **CUTSSL (OURS)** | **93.1** | **80.4** | **54.5** |

## C.2 Graph learning identifies hard samples

Table 7: Accuracy of different method on the top 5%, 10%, 20% of samples with the smallest margin.

| CIFAR-10 MARGIN THRESHOLD | 5% | 10% | 20% | 100% |
|---|---|---|---|---|
| EXACT LAPLACE/LP | 27.4 (4.3) | 29.6 (4.3) | 31.6 (4.1) | 41.6 (4.3) |
| POISSON | 29.1 (2.1) | 28.5 (1.6) | 30.1 (1.2) | 40.7 (5.5) |
| **CUTSSL (OURS)** | **36.2 (4.9)** | **43.2 (4.4)** | **44.5 (5.3)** | **44.7 (5.9)** |

Recall that Laplace learning solves a problem of the form

$$L_u X_{\text{lap}} = B := -L_{ul} Y.$$

It is then necessary to map the continuous-valued predictions given by $X_{\text{lap}}$ to elements in the label-set. Usually, the following post-hoc rounding step is performed:

$$\ell(i) = \arg \max_{j \in 1,...,k} (X_{\text{lap}})_{ij}.$$

However, there is one significant issue with this heuristic, as implied by proposition 4.1 in our paper. At low-label rates, $L_u$ is nearly singular and dominated by a large drift-term associated with the vector $L_u^{-1} \mathbf{1}_n$. This term is essentially a constant that is governed by the underlying graph and the distribution of the labeled vertices, but it can dominate any "useful" information and hinder the threshold heurstic. Notably, in the extreme case, thresholding in the low label rate regime results in one class dominating the predictions. This is presented in figure 5, which demonstrates that when the column-argmax heuristic is adopted a single class prediction dominates the rest.

This particular degeneracy is especially relevant for vertices near the decision boundary- e.g. vertices with corresponding rows in $X_{\text{lap}}$ that have small margin, where margin is defined

$$\text{margin}(i) = \max_{j \in [k]}(X_{\text{lap}})_{ij} - \max_{l \in [k], :l \neq j}(X_{\text{lap}})_{ij}$$

To illustrate this issue and how CutSSL addresses it, in Tab. 7 we evaluate Laplace and Poisson learning (methods that produce continuous-valued predictions) on the Cifar-10 dataset with one

sample labeled per class. We then consider the accuracy of each method on the top $5\%$, $10\%$, $20\%$ of samples with the smallest margin. We see that CutSSL, which makes discrete predictions, performs much better on samples with small margin.

Here, we provide examples of misclassified samples that lie on the "boundary" of their respective partitions. The node boundary of a set $V_1$ with respect to a set $V_2$ is the set of vertices $v$ in $V_1$ such that for some vertex $u$ in $V_2$, there is an edge joining $u$ to $v$. We show that the examples presented in Figure 5, particularly in the top row for MNIST, correspond to digits which are messily written or easily confused (bottom, Fashion-MNIST). We hypothesize that these challenging samples lie on the cut-boundary between partitions.

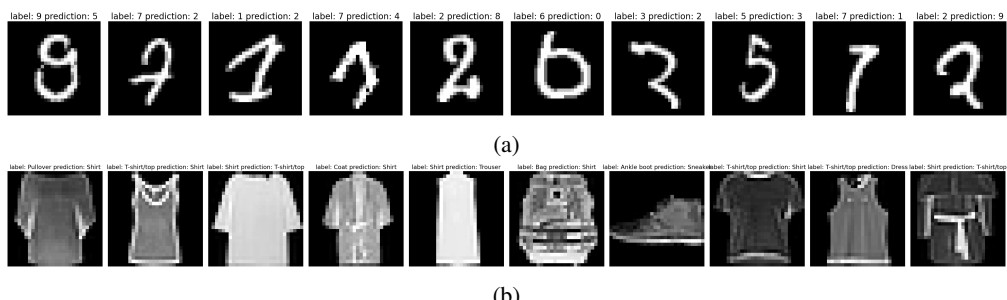

(a)

(b)

Figure 5: **(a)** Random sample of misclassified images on the cut-boundary on MNIST. **(b)** misclassified images on the cut-boundary of FashionMnist.

### C.3 Planetoid (Cora, Citeseer, Pubmed)

In the main text and earlier in the appendix, we provided numerical results on three popular image datasets. The graphs in those datasets are constructed as kNN graphs. In this section, we evaluate our method on non-kNN graphs corresponding to citation networks, where the graph is given. In these networks, the vertices represent documents and their links refer to citations between documents, while the label corresponds to the topic of the document. Due to the smaller size of these graphs, we set $s = 0.05$. We show in Fig. 2 that our method continues to exhibit significant improvement, beyond the state-of-the-art on these graphs.

In Table 8, we again outperform competing methods across all datasets and label rates. For example, on Citeseer at 100 labels per class, we outperform Poisson-MBO by $9.4\%$.

We omit results on Volume-MBO [19] due to runtime errors when evaluating the method on these graphs.

### C.4 Ablation study of $S$ and cut objective and graph construction

The two underlying motivations of our work are that (1.) the graph-based machine learning setting necessitates the assumption that class structure is tied to the cut structure of the graph (2.) searching for near-integer solutions when solving the semi-supervised partitioning problem is critical at low-label rates. We demonstrate this below by evaluating various choices of $S$ as well as evaluating the quality of the integer partitioning implied by the integer-rounded predictions made by various methods across datasets in Table 9.

Additionally, we discuss the choice of $S = sD$. $s$ is the primary hyperparameter of the CutSSL algorithm. As mentioned in the main text, our choice of $S$ is motivated from the perspective of regularization. A large enough choice of $s$ results in an indefinite or concave problem that has a combinatorial number of local minima. By choosing an appropriate sequence of $s$, we can control the number of poor local minima. In Figure 6(a) we demonstrate convergence with respect to the objective of (10) and (b) proximity to an integral solution with various choices of $s$. In Figure 7, we plot various measures associated with three different sequences $(s_0, s_1, \ldots, s_T)^2$, with $s_0 = 0$ and $s_T = 0.5$. Each $s_t$ is determined by $s_t = s_{t-1} + d$ for some fixed constant $d$. It can be seen that

---

[2]recall that our algorithm works by finding a solution to (10) given a particular $s_t$ and using this solution to initialize a subsequent problem associated with $s_{t+1}$

Table 8: Average accuracy over 100 trials evaluated on the largest connected subgraph with standard deviation in brackets. Best is bolded.

| CORA # LABELS PER CLASS | 1 | 3 | 5 | 10 | 100 |
|---|---|---|---|---|---|
| LAPLACE/LP [38] | 21.8 (14.3) | 37.6 (12.3) | 51.3 (11.9) | 66.9 (6.8) | 81.8 (1.1) |
| MEAN SHIFT LAPLACE/LP | 52.7 (7.6) | 63.3 (6.3) | 67.8 (4.8) | 72.1 (2.7) | 80.8 (1.1) |
| POISSON [10] | 59.8 (7.9) | 66.2 (5.8) | 72.4 (2.1) | 74.1 (1.8) | 79.8 (1.0) |
| POISSON-MBO [10] | 59.9 (6.4) | 69.1 (3.1) | 72.4 (2.4) | 74.3 (2.1) | 79.2 (0.9) |
| **CUTSSL (OURS)** | **67.4 (3.4)** | **73.2 (3.1)** | **75.8 (2.1)** | **78.7 (1.1)** | **85.4 (0.7)** |
| CITESEER | | | | | |
| LAPLACE/LP [38] | 27.9 (10.4) | 47.6 (8.1) | 56.0 (5.9) | 63.7 (3.5) | 71.6 (1.2) |
| MEAN SHIFT LAPLACE/LP | 56.8 (6.8) | 56.8 (6.9) | 60.3 (5.1) | 64.4 (3.4) | 70.6 (1.2) |
| POISSON [10] | 59.4 (5.4) | 59.4 (5.4) | 62.7 (4.2) | 66.9 (1.8) | 69.4 (1.0) |
| POISSON-MBO [10] | 47.7 (8.0) | 55.7 (3.2) | 61.0 (1.7) | 63.1 (1.7) | 67.0 (1.1) |
| **CUTSSL (OURS)** | **62.4 (4.6)** | **63.4 (7.2)** | **66.9 (1.4)** | **68.1 (1.3)** | **76.4 (0.9)** |
| PUBMED | | | | | |
| LAPLACE/LP [38] | 34.6 (8.8) | 35.7 (8.2) | 36.9 (8.1) | 39.6 (9.1) | 74.9 (3.6) |
| MEAN SHIFT LAPLACE/LP | 54.4 (11.1) | 62.7 (9.7) | 66.2 (8.5) | 69.7 (5.0) | 76.3 (1.1) |
| POISSON [10] | 56.7 (12.8) | 66.5 (6.6) | 68.4 (5.9) | 71.2 (3.4) | 75.8 (0.9) |
| POISSON-MBO [10] | 56.9 (7.3) | 67.9 (3.4) | 69.6 (3.1) | 71.4 (2.5) | 76.2 (0.8) |
| **CUTSSL (OURS)** | **63.1 (4.7)** | **70.4 (3.1)** | **72.8 (2.9)** | **74.1 (1.4)** | **78.3 (0.8)** |

Table 9: Average cut value, given by $tr(X^\top W X)$ over 100 trials evaluated with standard deviation in brackets. Units are $10^3$. Best is bolded.

| MNIST # LABELS PER CLASS | 1 | 3 | 5 | 10 |
|---|---|---|---|---|
| MEAN SHIFT LAPLACE/LP | 9.532 (141.1) | 9.549 (78.3) | 9.555 (49.0) | 9.561 (20.5) |
| POISSON [10] | 9.539 (152.2) | 9.553 (58.4) | 9.557 (33.3) | 9.560 (18.1) |
| POISSON-MBO [10] | 9.568 (62.5) | 9.571 (1.9) | 9.571 (3.2) | 9.571 (2.5) |
| **CUTSSL (OURS)** | **9.570 (35.8)** | **9.572 (5.0)** | **9.572 (10.9)** | **9.572 (1.3)** |
| FASHIONMNIST | | | | |
| MEAN SHIFT LAPLACE/LP | 103626.1 (234.3) | 10.355 (158.8) | 10.355 (149.5) | 10.359 (112.3) |
| POISSON [10] | 10.355 (173.2) | 10.352 (149.1) | 10.354 (145.7) | 10.356 (106.5) |
| POISSON-MBO [10] | 10.383 (71.6) | 10.387 (68.7) | 10.383 (77.3) | 10.385 (91.89) |
| **CUTSSL (OURS)** | **10.383 (71.2)** | **10.398 (70.5)** | **10.399 (69.5)** | **10.402 (42.5)** |
| CIFAR-10 | | | | |
| MEAN SHIFT LAPLACE/LP | 7.031 (275.7) | 7.010 (191.2) | 7.011 (199.9) | 7.010 (120.7) |
| POISSON [10] | 7.0 (169.5) | 6.987 (169.5) | 6.993 (138.5) | 6.995 (120.9) |
| POISSON-MBO [10] | 7.007 (113.8) | 7.020 (47.7) | 7.019 (39.2) | 7.022 (46.2) |
| **CUTSSL (OURS)** | **7.018 (47.6)** | **7.034 (40.9)** | **7.038 (37.8)** | **7.043 (17.1)** |

(1.) that $s$ need not be chosen to exactly satisfy the condition in Proposition. 3.2, i.e. may be chosen significantly *less* than the proposition implies to recover an integer solution. And (2.) our formulation is reasonably robust to the choice of the sequence of $s$, i.e. one only needs to select a few values of $s$ on which to run the algorithm ($d$ can be chosen quite large). As mentioned in the main text, for our experiments we use sequences of length 3 (i.e. 3 $s$).

In Table 10 we report the change in accuracy for various choices of the parameter $k$, used in the construction of the underlying $k$-NN graph. The change is reported with respect to $k = 10$ (the value used in results reported in the rest of the experiments). Mean and standard deviation are provided in parenthesis over 100 trials. We do observe some variation in the accuracy, however the results are mostly robust to different choices of $k$. Notably, the relative performance remains consistent (i.e.,

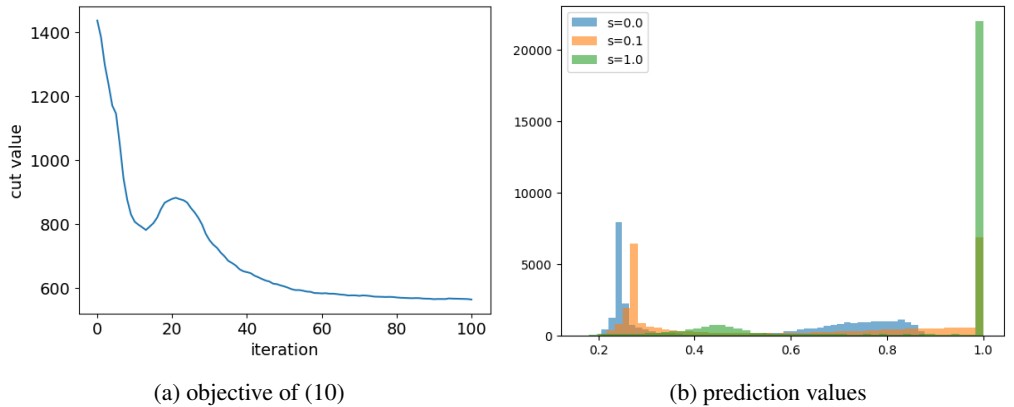

(a) objective of (10)                    (b) prediction values

Figure 6: **(a)** Cut value of class assignments with respect to $\mathcal{L}$. Strict descent is not guaranteed due to nonconvexity. **(b)** Distribution over $\max_i X_i$ for various choices of $s$ on Cifar-10. Larger $s$ leads to Boolean-valued solutions.

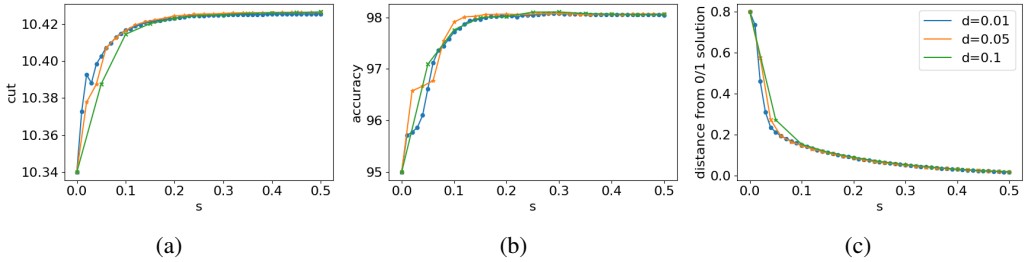

(a)                          (b)                          (c)

Figure 7: Comparison between different sequences of $s$ on MNIST. For various $s$ over three sequences, we plot the cut objective (a), accuracy (b), and proximity to integral solution (c).

CutSSL continues to outperform the other methods). One very interesting observation is that for CutSSL, the accuracy tends to improve with the value of $k$, although it's possible that the accuracy could deteriorate for larger values of $k$.

Table 10: Ablation study on neighborhood graph construction (choice of $k$).

| CIFAR-10 $k$: | 5 | 10 | 15 | 100 |
|---|---|---|---|---|
| EXACT LP | +1.00 (2.3) | 0 | -0.31 (2.1) | -0.47 (2.3) |
| POISSON | +0.30 (2.2) | 0 | -0.75 (2.1) | -1.0 (2.3) |
| POISSON-MBO | -0.32 (2.7) | 0 | -0.57 (2.9) | -2.30 (3.1) |
| **CUTSSL (OURS)** | -0.21 (2.3) | 0 | +0.16 (2.2) | +0.64 (2.1) |

In Table 11 we evaluate the effect mechanism used to construct the graph on MNIST. First, we evaluate two variants of weighting edges of the KNN-based method adopted in the main text. Singular refers to weights of the form $w_{ij} = 1/\|x_i - x_j\|_2$, where $x_i$ and $x_j$ are the features of the $i$-th and $j$-th data points. Uniform refers to $w_{ij} = 1$, i.e. an unweighted graph. In the third column, we evaluate a more sophisticated method for graph construction based on Non-Negative kernel regression (NNK) [28]. As opposed to the KNN-based methods, which only consider the distance (or similarity) between the query and the data, the NNK-based method takes into account the relative position of the neighbors themselves. Note that our CutSSL out-performs Poisson-MBO for the different graph weighting methods and NNK construction.

## D   Details of Previous work

In this section, we provide further details on two state of the art methods for volume-preserving label refinement.

Table 11: Ablation study on neighborhood graph construction (choice of $w_{ij}$).

| METHOD | KNN (GAUSSIAN) | KNN (SINGULAR) | KNN (UNIFORM) | NNK |
|---|---|---|---|---|
| EXACT LP | 91.0 (4.7) | 89.7 (2.8) | 89.8 (2.3) | 91.7 (1.1) |
| POISSON-MBO | 96.5 (2.6) | 94.2 (2.9) | 95.7 (1.6) | 97.3 (1.4) |
| **CUTSSL (OURS)** | 97.5 (0.1) | 95.8 (2.6) | 96.4 (2.0) | 98.4 (1.5) |

### D.1 PoissonMBO [10]

[10] propose a graph-cut method to incrementally adjust a seed decision boundary so as to improve the label accuracy and account for prior knowledge of class sizes. The proposed method applies several steps of gradient descent on the graph-cut problem:

$$\min_u \frac{1}{2}||\nabla u||^2_{\ell^2(X)^2} + \frac{1}{\tau}\sum_{i=1}^{n}\prod_{j=1}^{k}|u(x_i) - u_j|^2 \tag{35}$$

More concretely, the time-spitting scheme that alternates gradient descent on two energy functionals is employed:

$$E_1(u) := \frac{1}{2}||\nabla u||^2_{\ell^2(X)^2} - \mu\sum_{j=1}^{m}(y_k - \bar{y})\cdot u(x_j)$$
$$E_2(u) := \frac{1}{\tau}\sum_{i=1}^{n}\prod_{j=1}^{k}|u(x_i) - u_j|^2 \tag{36}$$

The first term $E_1$ corresponds to the Poisson learning objective. Gradient descent on the second term $E_2$, when $\tau > 0$ is small, amounts to projecting each $u(x_i) \in \mathbb{R}^k$ to the closest label vector $e_j \in S_k$, while preserving the volume constraint $(u)X = b$. This projection is approximated this by the following procedure: Let $Proj(S_k) : \mathbb{R}^k \to S_k$ be the closest point projection, let $s_1, \ldots, s_k > 0$ be positive weights, and set

$$u^{t+1}(x_i) = \text{Proj}_{S_k}(\text{diag}(s_1, \ldots, s_k)u^t(x_i)) \tag{37}$$

where $\text{diag}(s_1, \ldots, s_k)$ is the diagonal matrix with diagonal entries $s_1, \ldots, s_k$. We use a simple gradient descent scheme to choose the weights $s_1, \ldots, s_k > 0$ so that the volume constraint $(u^{t+1})X = b$ holds.

Note that Poisson MBO requires a fidelity parameter $\mu$, two parameters $N_{inner}$ and $N_{outer}$, and a time step parameter $\tau$, and additional clipping values $s_{min}$ and $s_{max}$.

### D.2 VolumeMBO [19]

The work of [19] provides an alternative way to enforce explicit class balance constraints with a volume constrained MBO method based on auction dynamics. Their method uses a graph-cut based approach with a Voronoi-cell based initialization.

More concretely, the solve a volume-constrained cut problem via successive linearizations. I.e., each iteration of the algorithm necessitates finding a solution to the following problem:

$$\max_Z -WZ$$
$$\text{s.t. } Z1 = 1, \ Z \geq 0, \ Z^\top 1 = m \tag{38}$$

## E  Algorithm derivations

Here, we describe algorithms referenced in the main text, including a projected CG method for solving (18) and a variant of the ADMM method introduced to solve (8) that incorporates a step size parameter.

## E.1 Detailed derivation of iterates $X$, $\mu_1$, $\mu_2$ in Sec. 3.2

Here, we provide the details on the derivations of the ADMM iterates and Lagrange multiplier updates $\mu_1$ and $\mu_2$.

First, define the Lagrangian, $G(X, T, \Lambda)$ to be the objective of the following problem:

$$\max_{\Lambda \in \mathbb{R}^{n \times k}} \min_{X, T \in \mathbb{R}^{n \times k}} \frac{1}{2} tr(X^\top L_{u,s} X) - tr(X^\top B)$$
$$+ tr(\Lambda^\top (X - T)) + \frac{1}{2} \| X - T \|^2 \tag{39}$$
$$\text{s.t. } X^\top \mathbf{1}_n = \mathbf{m}, \ X \mathbf{1}_k = \mathbf{1}_n, \ T \geq 0.$$

The first-order optimality conditions on $X$ are

$$0 = L_{u,s} X - B + \Lambda + (X - T) + \mathbf{1}_n \mu_1^\top + \mu_2 \mathbf{1}_k^\top$$
$$\implies X = \bar{L}_{u,s}^{-1}(B + T - \Lambda - \mathbf{1}_n \mu_1^\top - \mu_2 \mathbf{1}_k^\top), \tag{40}$$

where $\mu_1 \in \mathbb{R}^k$ and $\mu_2 \in \mathbb{R}^n$ are associated with the constraints $X^\top \mathbf{1}_n = \mathbf{m}$ and $X \mathbf{1}_k = \mathbf{1}_n$ respectively, and $\bar{L}_{u,s} = L_{u,s} + I$. The optimality conditions associated with $(\mu_1, \mu_2)$ are recovered by applying the constraint $X^\top \mathbf{1}_n = \mathbf{m}$.

$$\mathbf{m} = X^\top \mathbf{1}_n$$
$$= \{ \bar{L}_{u,s}^{-1}(B + T - \Lambda - \mathbf{1}_n \mu_1^\top - \mu_2 \mathbf{1}_k^\top) \}^\top \mathbf{1}_n \tag{41}$$
$$= (B + T - \Lambda - \mathbf{1}_n \mu_1^\top - \mu_2 \mathbf{1}_k^\top)^\top \bar{L}_{u,s}^{-1} \mathbf{1}_n.$$

Let $c = \mathbf{1}_n^\top \bar{L}_{u,s}^{-1} \mathbf{1}_n$.

Solving for $\mu_1$,

$$(B + T - \Lambda - \mu_2 \mathbf{1}_k^\top)^\top \bar{L}_{u,s}^{-1} \mathbf{1}_n - \mathbf{m} = \mu_1 c \tag{42}$$

and

$$\mu_1 = c^{-1} \{ (B + T - \Lambda - \mu_2 \mathbf{1}_k^\top)^\top \bar{L}_{u,s}^{-1} \mathbf{1}_n - \mathbf{m} \} \tag{43}$$

Similarly,

$$\mathbf{1}_n = X \mathbf{1}_k$$
$$= \{ \bar{L}_{u,s}^{-1}(B + T - \Lambda - \mathbf{1}_n \mu_1^\top - \mu_2 \mathbf{1}_k^\top) \} \mathbf{1}_k \tag{44}$$
$$= \bar{L}_{u,s}^{-1} \{ (B + T - \Lambda - \mathbf{1}_n \mu_1^\top - \mu_2 \mathbf{1}_k^\top) \} \mathbf{1}_k$$

Solving for $\mu_2$,

$$(B + T - \Lambda - \mathbf{1}_n \mu_1^\top) \mathbf{1}_k - \bar{L}_{u,s} \mathbf{1}_n = k \mu_2 \tag{45}$$

and

$$\mu_2 = \frac{1}{k} \{ (B + T - \Lambda - \mathbf{1}_n \mu_1^\top) \mathbf{1}_k - \bar{L}_{u,s} \mathbf{1}_n \} \tag{46}$$

## E.2 ADMM with adaptive step size for CutSSL

In this section, we introduce a variation of the algorithm presented in the main text. Here, we augment the algorithm with an additional tunable parameter $\beta$. This parameter offers flexibility depending on structure of the problem. In particular, the update to $X$ is characterized by a linear system with matrix $L_{u,s} + \beta I$. To ensure efficient computation of its inverse, $\beta$ needs to be chosen sufficiently large. However, the price of a large $\beta$ is convergence to an integer solution. This tradeoff needs to be carefully considered in the context of the graph and the label rate when selecting $\beta$. For all our experiments, we choose $\beta = 1$.

One may speed up the empirical rate of convergence of ADMM-type algorithms by integrating an adaptive step-size with the quadratic dual term in the augmented Lagrangian.

First, introduce multipliers $\mu_1 \in \mathbb{R}^k, \mu_2 \in \mathbb{R}^n, \nu \in \mathbb{R}^{n \times k}$ and recall the Lagrangian associated with (20).

$$\mathcal{G}(X, \mu_1, \mu_2, \nu) = \frac{1}{2} tr(X^\top L_{u,s} X) - tr(X^\top B) - \mu_2^\top(X \mathbf{1}_k - \mathbf{1}_n) - \mu_1^\top(X^\top \mathbf{1}_n - \mathbf{m}) - tr(\nu^\top X). \tag{47}$$

The KKT condition of $X$ in (47) is

$$\nu = L_{u,s}X - B - \mathbf{1}_n\mu_1^\top - \mu_2\mathbf{1}_k^\top \geq 0, \tag{48}$$

and $X_{ij} = 0$ must hold for those $(i, j)$ with $\nu_{ij} > 0$.

As mentioned in the main text, ADMM is carried out to reach a saddle point of $\max_\Lambda \min_{X,T} \mathcal{G}_\beta(X, T, \Lambda)$ where

$$\mathcal{G}(X, T, \Lambda) = \frac{1}{2}tr(X^\top L_{u,s}X) - tr(X^\top B) + tr(\Lambda^\top(X - T)) + \frac{\beta}{2}||X - T||_F^2 \tag{49}$$

subject to

$$T \geq 0, X\mathbf{1}_k = \mathbf{1}_n, X^\top\mathbf{1}_n = \mathbf{m}. \tag{50}$$

To implement the update of $X$, we observe that the optimality condition of $X$ is

$$L_{u,s}X - B + \Lambda + \beta(X - T) - \mathbf{1}_n\mu_1^\top - \mu_2\mathbf{1}_k^\top = 0 \tag{51}$$

Decompose $X$. $X = X_0 + X_1$, where $X_1$ is the fixed matrix $X_1 = n^{-1}\mathbf{1}_n\mathbf{m}^\top$. This decomposition implies that $X_0$ has sum-zero rows and columns, i.e. $X_0\mathbf{1}_k = 0$ and $X_0^\top\mathbf{1}_n = 0$. Hence, we can determine $X_0$ from

$$L_{u,s}X_0 + \beta X_0 = \mathbf{1}_n\mu_1^\top + \mu_2\mathbf{1}_k^\top - L_{u,s}X_1 - \beta X_1 + \beta T - \Lambda + B. \tag{52}$$

Apply the projections $P_1 = I - \mathbf{1}_n\mathbf{1}_n^\top/n$ and $P_2 = I - \mathbf{1}_k\mathbf{1}_k^\top/k$ to remove the column and row sums on both sides (to eliminate $\mu_1$ and $\mu_2$):

$$X_0 = (P_1L_{u,s}P_1 + \beta P_1)^\dagger P_1(-L_{u,s}X_1 - \beta X_1 + \beta T - \Lambda + B)P_2, \tag{53}$$

Next, the update of $T$ and $\Lambda$ proceed as before:

$$T \leftarrow \max(\beta^{-1}\Lambda + X, 0), \quad \Lambda \leftarrow \Lambda + \beta(X - T) \tag{54}$$

Once ADMM converges, i.e., $||X - T|| \to 0$, according to (54) and $T \geq 0$, we have that $\Lambda_{ij} \leq 0$ and $\Lambda_{ij} = 0$ holds for $X_{ij} > 0$. Additionally, by (51), we have

$$L_{u,s}X - B - \mathbf{1}_n\mu_1^\top - \mu_2\mathbf{1}_k^\top = -\Lambda \geq 0, \tag{55}$$

i.e., we reach a KKT point associated with the condition (48).

### E.3 Projected Conjugate Gradient for Laplace learning with cardinality constraints

Consider the problem

$$\min_{X \in \mathbb{R}^{n \times k}} \left\{ f(X) = \frac{1}{2}tr(X^\top L_uX) - tr(X^\top B) \right\} \tag{56}$$
$$\text{s.t. } X^\top\mathbf{1}_n = \mathbf{m}$$

In practice, large-scale Laplacian linear systems are typically solved using preconditioned conjugate gradient. Given the associated equality-constrained problem

$$\min_X \frac{1}{2}\langle X, L_uX \rangle - \langle X, B \rangle \tag{57}$$
$$\text{s.t. } X^\top\mathbf{1}_n = \mathbf{m}$$

One practical modification of conjugate gradient involves first choosing a feasible initialization $X_0$ satisfying $X^\top\mathbf{1}_n = \mathbf{m}$ and running *projected conjugate gradient* Chapter 16.3 [27], ensuring that intermediate residuals—update directions—lie in the orthogonal complement of $\mathbf{1}_n$. Termination is determined according to the residual.

# F   Limitations and Broader Impact

Here, we list several limitations and briefly touch upon the broader societal impact of our method.

- Underlying graph structure: Much of this work relies on the structure of the underlying graph. In this work, we explored the applicability of our method to $k$-NN graphs, generated from neural network-derived features and citation networks.
- The choice of $S = sD$ is one heuristic that satisfies the conditions outlined in Hager's paper [17]. We are currently exploring the effect of different choices of $S$ on the convergence and overall efficacy of our method. Additionally, it is interesting to consider how the graph structure, number of unlabeled and labeled nodes might inform better choices of $S$.

Regarding the broader societal impact of our method, we have introduced a new method for graph learning that improves on the state of the art across multiple label-rate regimes. The development of such methods has positive and negative ramifications for civil liberties and human rights. While the positive ramifications are relatively clear—models which require less data are cheaper to train and are generally "lower-impact"—the negative consequences are not immediately obvious. The above question is explored in greater detail in the context of semi-supervised graph learning in Fairness Zhang et al. [35]. The principal claim made by Zhang et al. [35] is that when quantities of unlabeled data are provided, label propagation may be modified to improve fairness.

