# OpenReview forum: "Continuous Partitioning for Graph-Based Semi-Supervised Learning"
_NeurIPS.cc/2024/Conference — NeurIPS 2024 poster_

### Official Review · Reviewer_8RfG · 2024-07-10

**Soundness:** 2
**Presentation:** 1
**Contribution:** 2
**Rating:** 4
**Confidence:** 3

**Summary:**

In this submission, authors proposed a framework for graph-based semi-supervised learning based on continuous nonconvex quadratic programming.

**Strengths:**

This paper studies the graph-based semi-supervised learning problem, which has attracted many attentions.  In this submission, authors proposed a framework for graph-based semi-supervised learning based on continuous nonconvex quadratic programming. Experiments show that the proposed CutSSL framework significantly surpasses the current state-of-the-art on k-nearest neighbor graphs and large real-world graph benchmarks.

**Weaknesses:**

There are many problems in this article. The most problem is that the motivation and contribution are unclear. At the beginning of abstract, authors argue that Laplace learning algorithms for graph-based semi-supervised learning have been shown to suffer from degeneracy at low label rates and in imbalanced class regimes. Then, they propose the CutSSL, a framework for graph-based semi-supervised learning based on continuous nonconvex quadratic programming. Although experiments validate the superiority of the proposed CutSSL against competing method across a variety of label rates, class imbalance, and label imbalance regimes (by the way, what is the difference between class imbalance and label imbalance), I cannot find the relationship between the motivation and the technical design of CutSSL. Moreover, the meaning of the term "Continuous Partitioning" in title is also unclear. For the proposed CutSSL, according to the summarized contributions at the end of Section 1, the main contribution of CutSSL is the mathematical framework and its ADMM-based optimization algorithm. These contributions have little to do with the aforementioned motivation of this paper. From a technical point of view, the main contribution of the CutSSL is Eq.(9), where S=sD. Then, another contribution is how to optimize this formulation, which is presented in Section 3.2. Note that in Eq.(10), there are only two matrix parameters to be optimized (i.e., X and T), but it is hard to find throughout this paper that how to determine s and B. Similar problems exist in many places. Ref.[16] is the main reference of this paper, but do not rely on the notations or backgrounds in it too much. Besides, there is a recent review named ''Graph-Based Semi-Supervised Learning: A Comprehensive Review'' published in TNNLS is not mentioned in this paper, which means a literature review should be carefully conducted.

**Questions:**

Please clarify the motivation and contribution problem mentioned above.

**Limitations:**

Some limiattions are discussed in appendix.

---

> ### Author Rebuttal · Authors · 2024-08-07
>
> We thank the reviewer for their constructive comments. We address reviewer comments below and will edit the final version of the paper according to the reviewer’s comments.
>
> ***Motivation of CutSSL***
> In contrast to Laplace learning, which is degenerate in the low-label rate regime, we start with a framework that is nondegenerate, even without any labels (min cut, stated in eq (4)) and derive a nonconvex continuous relaxation that incorporates label information. To show the relaxation is exact (eq (9)), we prove two statements (propositions 3.1 and 3.2). First (proposition 3.1), we show that at integer solutions (valid cuts), the relaxation coincides with the combinatorial problem, i.e., the min cut with respect to the relaxation is the min cut with respect to the combinatorial problem. Next (proposition 3.2), we show that (1.) local minima of the relaxation are integer-valued and (2.) the number of local minima can be controlled (and is via our framework) to prevent a badly behaved optimization problem with many spurious local minima. Thus the global minimum of the relaxation is exactly the min cut.
>
> To provide additional motivation for CutSSL, we provide the experiment below. We show that in the absence of labels purely unsupervised graph cut-based methods (e.g. Spectral Clustering (SC)) yield a graph cut / partition such that the resulting clusters are well-aligned with the labels. To demonstrate this, we calculate a partition using both SC and a graph cut solution to equation 8 (‘cut’), i.e.this is CutSSL in the absence of labels. To evaluate the alignment of the clusters yielded by the cut with the labels, we solve a linear sum assignment problem using the true labels to find the best permutation of cluster assignments. This assignment of clusters to classes shows that  the graphs cut are relatively well-aligned with the class labels in contrast to Laplace learning (LP) which degenerates in the low label rate regime (i.e. just a single label per-class). Thus, cut-based techniques work since they are derived from a fundamentally non-degenerate framework and our CutSSL augments the multi-way graph partition with labeled nodes.
>
>
> | Method    | MNIST | F-MNIST |  CIFAR10 |
> | -------- | ------- |------- |------- |
> | Laplace/LP |   16.1 (6.2)  | 18.4 (7.3)    | 10.4 (1.3)    |
> | CutSSL    | 97.5 (0.1)   |  64.3 (4.6)   | 44.7 (5.9)   |
> | Cut    |  95.9 (0.0)  |  62.7 (0.0)   | 41.1 (0.0)   |
> | SC    | 92.6  (0.0) |   60.5 (0.0)  |  35.3 (0.0)  |
>
> ***Clarification of the term Continuous Partitioning***
> Thank you for pointing out this issue. We will clarify this term in the final version. To be precise, we design a continuous relaxation of the combinatorial partitioning problem with label constraints. This term is typically used in the numerical optimization and partitioning community [1, 2], but is perhaps not as well-known in a machine learning context. We will clarify this term in the final version.
>
> [1] Pardalos, Continuous Approaches to Discrete Optimization Problems, Nonlinear Optimization and Applications, 1996.
> [2] Discrete Optimization via Continuous Relaxation, Symposium on Bridging Continuous and Discrete Optimization
>
> ***Clarification of the class and label imbalance and notation***
> To clarify, an imbalance in the class distribution refers to an imbalance in the number of samples for each class. A label imbalance refers to an imbalance in the number of samples that are labeled for each class. Note that the expression for the variable $B$ is given in equation (3).
>
> ***Choice of $s$***
> Thanks for pointing this out. This is an important question to address and we will provide additional clarification in the final version of our paper. We highlight a result that appears in the main text and several experiments presented in the appendix of our paper. In particular, the choice of $s$ is governed by Proposition 3.2, which provides an upper bound on the choice of $s$ that guarantees recovery of integer-valued solutions (for intuition see figure 1). A trivial lower-bound is given by one convex relaxation of the problem ($s = 0$). Given the upper and lower-bound, we construct a homotopy path, a sequence of solutions $X(s)$ that vary in $s$. In practice, we choose a linear or geometric sequence s between $0$ and the upper bound. Many times, the upper bound need not be reached to recover a solution that is very close to integer (see Figure 6.b in the appendix). Likewise, we demonstrate that in practice, one can take a very coarse sequence of $s$. This result is presented and discussed in Figure 6 in appendix C.5. In fact, to reproduce the results in the main text, only a single nonzero data-dependent value of s is used (i.e. the homotopy path consists of constructing solutions for $s = 0$ and $s = 0.1$).
>
>
> ***Additional references***
> Thanks for notifying us of this missing review. We have added it to our paper. It will appear in the final version.

---

> > ### Comment · Reviewer_8RfG · 2024-08-11
> >
> > Thanks for your response, some of my concerns are clarified. I am open to hear discussions from other reviewers.

---

> > > ### Author Response · Authors · 2024-08-12
> > >
> > > Thanks for investing time and for providing valuable feedback! We appreciate the suggestions and your help with improving the quality of our paper. Having two days left for the discussion period, we would be happy to run additional experiments, address any remaining issues, or clarify sources of confusion, if there are any. If our responses have adequately addressed the concerns, we kindly ask the reviewer to consider raising our score.

---

### Official Review · Reviewer_aiM6 · 2024-07-12

**Soundness:** 3
**Presentation:** 3
**Contribution:** 3
**Rating:** 8
**Confidence:** 4

**Summary:**

The paper considers the task of semi-supervised learning under cardinality constraints and proposes a framework based on a reformulation into a non-convex constrained quadratic program. The authors provide sufficient conditions for the exact recovery of integer solutions. Moreover, they present an algorithm called CutSSL which is based on the ADMM method to optimize the non-convex objective, which is shown to converge to a KKT point of their objective.

The paper also shows the connection of their framework to Laplace Learning, which provides an explanation for the efficacy of the commonly used mean-shift heuristic. The method is evaluated on three image datasets, citation networks as well as other large scale networks. The proposed approach consistently achieves superior results in terms of accuracy compared to the competing methods, while being in the same order of magnitude in terms of runtime compared to the fastest method.

**Strengths:**

The paper proposes a new graph-based framework as well as an iterative algorithm which is widely applicable.  In the experiments, the approach is shown to consistently outperform state-of-the-art methods on a wide range of datasets including image datasets and large scale networks.

**Weaknesses:**

It was not clear to me how exactly the sequence of s values is chosen. In the experimental section it says that s=0.1. I assume this is a typo since it would mean that s is kept fixed after the first step. What is meant here, the maximal s value (denoted s_T in the appendix), or the difference between s values (denoted d in the appendix)? Also, is the sequence of s values chosen in advance or is there some criteria to stop increasing the s values?

**Questions:**

See weaknesses section.

**Limitations:**

The authors discuss limitations of the approach, regarding the dependence on the choice of underlying graph structure, in the appendix.

---

> ### Author Rebuttal · Authors · 2024-08-07
>
> We thank the reviewer for their positive comments. Below we comment on the choice of the hyperparameter $s$.
>
> ***Choice of s***
> Thanks for pointing this out. This is an important question to address and we will provide additional clarification in the final version of our paper. We highlight a result that appears in the main text and several experiments presented in the appendix of our paper. In particular, the choice of $s$ is governed by Proposition 3.2, which provides an upper bound on the choice of $s_T$ that guarantees recovery of integer-valued solutions (for intuition see figure 1). A trivial lower-bound is given by one convex relaxation of the problem ($s_0 = 0$). Given the upper and lower-bound, we construct a homotopy path, a sequence of solutions $X(s_t)$ that vary in $s_t$. In practice, we choose a linear or geometric sequence s between $0$ and the upper bound (e.g. determined by that $d$ parameter). Many times, the upper bound need not be reached to recover a solution that is very close to integer (see Figure 6.b in the appendix). Likewise, we demonstrate that in practice, one can take a very coarse sequence of $s$. This result is presented and discussed in Figure 6 in appendix C.5. In fact, to reproduce the results in the main text, only a single nonzero data-dependent value of s is used (i.e. the homotopy path used for the main experiments consists of constructing a pair of solutions for $s = 0$ and $s = 0.1$).
>
> Generally, convergence can be measured according to proximity to an integer solution. E.g., if $X(s_t)$ is the CutSSL solution with respect to $s_t$, let $Proj_B(X(s_t))$ be the projection onto the set of feasible binary matrices (those binary matrices with one "1" in each row and with column sum $m$). Then, convergence has been reached if $||X(s_t) - Proj_B(X(s_t))||_F$ is sufficiently small.

---

> > ### Comment · Reviewer_aiM6 · 2024-08-12
> > **Response to Rebuttal**
> >
> > In the rebuttal, the authors provide additional details regarding the choice of the parameter s, as requested in my review. Thank you for your response.

---

### Official Review · Reviewer_Gcf5 · 2024-07-13

**Soundness:** 2
**Presentation:** 2
**Contribution:** 2
**Rating:** 6
**Confidence:** 4

**Summary:**

The paper proposes an approach for graph-based semi-supervised learning which is based on a cardinality-constrained extension of the classical Laplacian learning approach due to Zhu et al. Known theoretical limitations of standard Laplacian in the low-label rate regime motivates the approach. Theoretical properties of the approach are studied, including a sufficient condition for avoiding the need for rounding solutions and a practical optimization technique is provided for the cardinality-constrained Laplacian objective. Experiments demonstrate empirical advantage in the low-label rate regime without deterioration in standard regimes, and gains in the class imbalance regime. A connection is established to the mean-shift heuristic.

**Strengths:**

- The paper studies the fundamental problem of semi-supervised learning in important low label rate and class imbalance regimes, which have important practical consequences (excess unlabeled data does not become a disadvantage; fairness).
- The overall approach is a principled extension of Laplace learning to cardinality constraints with optimization techniques for the resulting non-convex objective.
- The experiments indicate practical usefulness of proposed approach.

**Weaknesses:**

- There is limited theoretical evidence for the claims that the proposed cardinality-constrained graph partitioning approach resolves the issues of low label rate and class imbalance.
- $s$ seems to be an important hyperparameter in the approach but is set to a constant value 0.1 without adequate justification.
- The connection to mean-shifted Laplace learning seems somewhat unclear. One, it is not clear if solution for (17) is close to CutSSL. Second the guarantees seem to be weak in the low-label rate regime (the bound connecting the two methods diverges in the regime of interest).
- I find it surprising that label/class imbalance experiments are relegated to the appendix despite being a repeated claimed contribution, and also there does not seem to be theoretical evidence for better performance in this regime either.

-----------------------------------------------------------------------------------------------------------------------------------------------------------
Post-rebuttal: Some of the above concerns (role of $s$ and connection to mean-shifted Laplace learning) have been addressed in the rebuttal and I have increased my score accordingly.

**Questions:**

- How can the cardinality constraints for different classes be known/estimated in practice?
- Line 28: "thresholding can further exacerbate the aforementioned degeneracy." Is there a reference or theoretical justification for this? Further elaboration on how thresholding exacerbates degeneracy would be helpful.
- $0 − 1$ minimizer, solution, assignment $\rightarrow$ $0$-$1$
- Line 39-40, it would be good to summarize what the connection is.
- What is the difference (if any) between formulations (8) and (9)?
- Is the (smallest) $s$ satisfying Proposition 3.2 easy to compute practically?
- Line 195: "The convergence of the standard two-block ADMM for convex and nonconvex problems has been thoroughly established in the literature". What is the relevant convergence rate for the current non convex problem?
- Line 232: "Our analysis of this problem provides new evidence to explain the empirical performance of this heuristic, while also revealing that it is suboptimal in some sense." Can you elaborate on this? Proposition 4.1 seems to imply that solutions of mean-shift heuristic are close to cardinality constrained Laplace learning (17).
- Reference [8] is incomplete.

**Limitations:**

The authors note that a rigorous theoretical analysis of exact cut-based methods is missing and note more limitations in a dedicated section in the appendix.

---

> ### Author Rebuttal · Authors · 2024-08-07
>
> We thank the reviewer for their careful reading of our work and for their positive and constructive comments and for appreciating the significance of our work. We address reviewer questions and suggestions below.
>
> ***Choice of S***
> We refer to the response to all authors. In summary, Proposition 3.2 provides an upper bound on the choice of $s$ that guarantees recovery of integer-valued solutions. A lower-bound is given by one convex relaxation ($s = 0$). Given the upper and lower-bound, we construct a homotopy path, a sequence of solutions $X(s)$ that vary in $s$. Empirically, the upper bound need not be reached to recover a solution that is very close to integer and that in practice, one can take a very coarse sequence (Figure 6 in appendix C.5.)
>
> ***Connection to mean-shift Laplace learning and Proposition 4.1***
> This is an interesting and important question. We agree that a more careful analysis needs to be conducted to derive conditions on the parameters of the problem for the solutions to (17) and cutSSL for $s=0$ to be close (we “show” this experimentally, see the paragraph below (17)). To summarize, we derive a connection between a special case of our framework and Laplace learning (proposition 4.1). This connection bridges our method with existing literature on Laplace learning. In particular, note that mean-shift Laplace learning is a reasonable approximation of CutSSL only in the case where $s \approx 0$. When $s$ is sufficiently large, the approximation does not hold. The indefiniteness of the quadratic yields an unbounded problem. We will clarify this in the final version of the paper.
>
> Regarding the bound, we believe that the fact that the approximation does not hold to be of interest. The purpose of the bound is to explore an existing way to address the degeneracy of Laplace learning: the mean-shifted Laplace learning. Our perspective of this algorithm as a heuristic for solving (17) results in (1.) a characterization of the degeneracy as a rank-1 perturbation of the solution to (17) and (2.) a simple improvement on the mean-shift heuristic. Note that the degenerate term (equation 32 in appendix B) grows as the amount of labeled data diminishes resulting in a larger gap between (17) and mean-shift laplace learning.
>
> ***Label/class imbalance performance***
> We agree that investigating guarantees for label and class imbalance performance is an important future direction. The main barrier to developing these theoretical results is nonconvexity and a dependence on the graph, the underlying labeling of the vertices, and the distribution of provided labels. In the future we plan to explore equivalent reparameterizations of the problem, e.g. a convex objective over a nonconvex set.
> We agree with the reviewer’s comment on the placement of the experiments for label/class imbalance and we will move these results to the main text in the final version of the paper.
>
> ***Additional questions***
> We thank the reviewer for carefully reading our work. We appreciate the detailed questions. We will add additional clarification to the final version of the paper.
>
> _Estimation of the cardinality constraint:_ In many situations, a prior may be obtained on the class distribution (e.g. using the provided labels). However, in general these may not necessarily be known. In this case, a typical assumption is that the classes are balanced (i.e. a uniform cardinality constraint). In the case that one does not have exact estimate of the prior, it would be interesting to investigate a variation of our problem which alters the cardinality constraint with upper and lower bounds: i.e. $ m - \epsilon<= X^\top 1_n <= m + \epsilon$.
>
> _Thresholding Laplace learning:_ In our paper we provide an alternative characterization to the degeneracy in prop 4.1 that was originally rigorously studied in (Calder et al., Poisson Learning: Graph Based Semi-Supervised Learning At Very Low Label Rates, 2020). This degeneracy results in predictions that are constant at low label rates. We should clarify that there are several ad-hoc heuristics that can be applied to map continuous valued predictions to discrete predictions, although thresholding is almost always the default choice. Although it is possible that alternative heuristics might perform better in the degenerate regime, we show specifically that thresholding is a poor choice (see figure 4 in the appendix) as it tends to make constant predictions.
>
> _Connection to spectral graph theory:_ Thanks for pointing this out. This is an important question to address. At a high level, both spectral partitioning methods and our method rely on two different continuous relaxations of the combinatorial graph partitioning problem (4). In fact, the set of valid cuts can exactly be characterized as the intersection of two sets: the feasible set of (8) (this is the one we do optimization over) and the set of orthogonal matrices. Informally, problem (5) is related to finding the binary matrix satisfying the cardinality constraint whose projection onto the orthogonal complement of 1 is closest to the eigenspace of the smallest nonzero eigenvalue of graph Laplacian.
>
> _(8) and (9):_ They are equivalent by proposition 3.1.
>
> _Smallest choice of $s$:_ Given the graph Laplacian $L = D - W$, the smallest $s$ satisfies the conditions for prop 3.1 is given by the expression $(s-1)(D_{ii}+D_{jj})  \geq 2w_{ij})$. One can compute the smallest $s$ analytically by computing the expression $1 + \max_{ij}2w_{ij} / (D_{ii} + D_{jj})$. This expression involves a simple element wise division and scan, taking $O(|E|)$ linear time in the number of edges of the graph / nonzero entries of $W$.
>
> _Convergence rates for ADMM:_ we clarify that while global convergence is thoroughly established, the convergence rate of ADMM for nonconvex problems is an ongoing research area. Under certain convexity and smoothness assumptions, ADMM converges linearly which we expect to hold for our problem.

---

> ### Author Response · Authors · 2024-08-13
>
> Thanks for investing time and for providing valuable feedback! We appreciate the suggestions and questions and your help with improving the quality of our paper. To follow-up on the choice of $s=0.1$, we compute the smallest values of $s$ that guarantee integer solutions on MNIST, FashionMNIST, and CIFAR10 (using the equation provided in the rebuttal). We will provide a more detailed experiment in the final version of the paper to explore the effect of the labeling.
>
> |   MNIST    | FashionMNIST | CIFAR10 |
> | ----------- | ----------- |----------- |
> | $s = 0.6$      | $s = 0.7$       | $s = 1.1$ |
>
> It can be seen that these values are approximately on the order of $0.1$ (the value that we adopt in the main table in the paper) and agree with the results provided in the appendix (Figures 6b and 7c). Note that we also evaluate larger values of s, (i.e. s=0.5, 1.0, etc.) in the appendix. In practice, we note that (1.) generally a choice of $s$ that is smaller than the bound yields solutions close enough to integer and (2.) the effect of the concave term (i.e. larger $s$ inducing more bad local minima) and the computational overhead of running CutSSL on larger sequences of $s$ need to be considered. This tradeoff is explored in Figure 6b in the appendix and these results (notably the robustness of CutSSL to coarse sequences of $s$ and $s$ smaller than the bound) motivated our choice of $s=0.1$ for the results presented in the main text. We will clarify this in the final version of the paper.
>
> Thank you again for the high-quality review. We would be happy to address any remaining issues, or clarify sources of confusion.

---

### Official Review · Reviewer_5yWx · 2024-07-22

**Soundness:** 2
**Presentation:** 3
**Contribution:** 2
**Rating:** 5
**Confidence:** 4

**Summary:**

The paper introduces CutSSL, a novel framework for graph-based semi-supervised learning (SSL). The authors address the limitations of Laplace learning algorithms, particularly their poor performance at low label rates and in imbalanced class scenarios. CutSSL leverages continuous nonconvex quadratic programming to achieve integer solutions, inspired by a cardinality-constrained minimum-cut graph partitioning problem. The framework utilizes the ADMM for robust and efficient problem-solving.

**Strengths:**

Originality: The paper introduces a fresh approach to graph-based SSL by framing it as a continuous nonconvex quadratic programming problem, a novel perspective compared to traditional Laplace learning methods.

Quality: The theoretical formulation is robust, and the use of ADMM for solving the quadratic programs is well-justified, offering solid convergence guarantees.

Clarity: The paper is well-structured and clearly explains the motivation, methodology, and contributions. The connection between their method and existing heuristics like mean-shifted Laplace learning is particularly insightful.

Significance: By addressing the degeneracy of Laplace learning at low label rates and in imbalanced settings, the paper tackles a significant challenge in SSL. The performance improvements on various benchmark datasets highlight the practical relevance of their approach.

**Weaknesses:**

Theoretical analysis: The paper would be better if more theoretical analysis could be provided on why the proposed method works when the label rate is extremely low or unbalanced. It is quite unclear on the theoretical motivation behind this proposed method.

Experimental Validation: While the results are promising, the experimental setup could benefit from a more extensive comparison with a broader set of baseline methods. Including more recent graph-based SSL algorithms, especially those GNN-based methods would strengthen the empirical validation.

Scalability: Although the authors claim that their method scales well to large graphs, a detailed analysis of the actual runtime comparisons with other methods would be beneficial.

Parameter Sensitivity: The paper does not discuss the sensitivity of CutSSL to its hyperparameters. An analysis of how different parameter settings affect performance would help in understanding the robustness of the method.

**Questions:**

1. Can you provide some theoretical analysis on why the proposed method works when the label rate is low or unbalanced?

2. How do recent GNN-based methods perform when compared with your proposed method?

3. How long will the proposed method run on some ogbn datasets you used in the experiments?

4. How sensitive is CutSSL to its hyperparameters? Could you provide an analysis or guidelines on setting these parameters?

5. How does CutSSL perform on different types of graph construction methods beyond the kNN graphs tested in the paper, still using the MNIST datasets instead of the real-world graph datasets?

**Limitations:**

The authors have addressed some limitations of prior methods, particularly the degeneracy of Laplace learning at low label rates and in imbalanced class scenarios. However, the paper could benefit from a more detailed discussion of the following:

Running Time Overhead: The continuous nonconvex quadratic programming approach may introduce computational overhead. A comparison of actual computational costs with other methods would be useful.

---

> ### Author Rebuttal · Authors · 2024-08-07
>
> We thank the reviewer for their positive and constructive comments and for appreciating the significance and originality of our work. We address reviewer questions and suggestions below.
>
> ***Theoretical analysis in low-label rate regime***
> We agree that this is an interesting and important direction. To be clear, In this paper, we make three formal mathematical statements. In contrast to Laplace learning, which is degenerate in the low-label rate regime, we start with a framework that is nondegenerate, even without any labels, but combinatorial problem (min cut, stated in eq (4)) and derive a nonconvex continuous relaxation that incorporates label information. To show the relaxation is exact (eq (9)), we prove two statements (propositions 3.1 and 3.2). First (proposition 3.1), we show that at integer solutions (valid cuts), the relaxation coincides with the combinatorial problem, i.e., the min cut with respect to the relaxation is the min cut with respect to the combinatorial problem. Next (proposition 3.2), we show that (1.) local minima of the relaxation are integer-valued and (2.) the number of local minima can be controlled (and is via our framework) to prevent a badly behaved optimization problem with many spurious local minima. Thus the global minimum of the relaxation is exactly the min cut.
>
> However, a precise statement about the quality of the solutions recovered by CutSSL is beyond the scope of this work. Such a statement would need to take into account the topology of the network, the true labeling of all the vertices, and the distribution of the provided labels and additionally address the underlying nonconvexity of the problem. In a future work we plan to explore equivalent reparameterizations of the problem, e.g. a convex objective over a nonconvex set.
>
> ***GNN-based methods***
> In tables 2 and 8 we included a comparison to a recent GNN-based method designed for low-label rate semi-supervised learning (GraphHop, Xie et al. 2023). We also compare to SOTA method Poisson MBO (Calder et al. 2020) and show we out-perform both methods, with ~30% improvement of CutSSL compared to GraphHop. We will include comparison to additional GNN-based methods in the final version.
>
> ***Runtime analysis***
> We highlight additional results that appear in the main text and appendix of our paper. In particular, in table 2 of section 5 of the main text and Table 8 in Appendix C4 we provide the average runtime over 100 trials on the large-scale OGB-Arxiv and OGBN-Products benchmarks. While a marginal increase in computation cost of CutSSL is apparent, when compared to classic Laplace and Poisson learning methods, we emphasize that the cost is small and is vastly superior compared to PoissonMBO and the GNN-based method GraphHop. The price of each iteration of ADMM is very cheap due the well-conditioned nature of the subproblems (see section 3.3 for more details).
>
> ***Choice of hyperparameter / sensitivity***
> We highlight a result that appears in the main text and several experiments presented in the appendix of our paper. In particular, the choice of $s$ is governed by Proposition 3.2, which provides an upper bound on the choice of $s$ that guarantees recovery of integer-valued solutions (for intuition see figure 1). A trivial lower-bound is given by one convex relaxation of the problem ($s = 0$). Given the upper and lower-bound, we construct a homotopy path, a sequence of solutions $X(s)$ that vary in $s$. In practice, we choose a linear or geometric sequence s between $0$ and the upper bound. Many times, the upper bound need not be reached to recover a solution that is very close to integer (see Figure 6.b in the appendix). Likewise, we demonstrate that in practice, one can take a very coarse sequence of $s$. This result is presented and discussed in Figure 6 in appendix C.5. In fact, to reproduce the results in the main text, only a single nonzero data-dependent value of s is used (i.e. the homotopy path consists of constructing solutions for $s = 0$ and $s = 0.1$).
>
> ***Beyond KNN-Graphs***
> Thanks for this suggestion. This is an interesting direction for future work. We note that in Table 10 in appendix C.5, we provide an ablation for various choices of k. As mentioned in the appendix, the relative performance remains consistent (i.e., CutSSL continues to outperform the other methods). One very interesting observation is that for CutSSL, the accuracy tends to improve with the value of $k$, although it’s possible that the accuracy could deteriorate for larger values of $k$.
> In the table below we evaluate the effect mechanism used to construct the graph on MNIST. First, we evaluate two variants of weighting edges of the KNN-based method adopted in the main text. Singular refers to weights of the form $w_{ij} = 1/||x_i - x_j||$, where $x_i$ and $x_j$ are the features of the $i$-th and $j$-th data points. Uniform refers to $w_{ij} = 1$, i.e. an unweighted graph. In the third column, we evaluate a more sophisticated method for graph construction based on Non-Negative kernel regression (NNK) [1]. As opposed to the KNN-based methods, which only consider the distance (or similarity) between the query and the data, the NNK-based method takes into account the relative position of the neighbors themselves. Note that our CutSSL out-performs Poisson-MBO for the different graph weighting methods and NNK construction.
> We thank the reviewer for this suggestion and we will add this experiment to the supplement in the final version of the paper.
>
> | Method    | KNN (Gaussian) | KNN (Singular) |KNN (Uniform) |NNK |
> | -------- | ------- |------- |------- |------- |
> | Mean shift Laplace | 91.0 (4.7)    |  89.7 (2.8)  | 89.8 (2.3)    | 91.7 (1.1)    |
> | PoissonMBO | 96.5 (2.6) | 94.2 (2.9) | 95.7 (1.6) | 97.3 (1.4)  |
> | CutSSL    | 97.5 (0.1)    |  95.8 (2.6)    | 96.4 (2.0)    | 98.4 (1.5)    |
>
> [1] Shekkizhar and Ortega, Neighborhood and Graph Constructions using Non-Negative Kernel Regression, ICASSP, 2020

---

### Author Rebuttal · Authors · 2024-08-07

We thank all reviewers for carefully reading our work and for their constructive suggestions and comments. We have added individual responses and several additional experiments suggested by the reviewers. We will incorporate these in the final version of the paper. Below, we summarize the main contributions of the paper and expand on the choice of s parameter and introduce two new experiments based on discussions with the reviewers.

***Summary of contributions***
In contrast to Laplace learning, which is degenerate in the low-label rate regime, we consider a framework that is nondegenerate, even without any labels (the minimum cut problem stated in eq (4)) and derive a nonconvex continuous relaxation that incorporates label information. To show the relaxation is exact (eq (9)), we prove two statements (propositions 3.1 and 3.2). First (proposition 3.1), we show that at integer solutions (valid cuts), the relaxation coincides with the combinatorial problem, i.e., the min cut with respect to the relaxation is the min cut with respect to the combinatorial problem. Next (proposition 3.2), we show that (1.) local minima of the relaxation are integer-valued and (2.) the number of local minima can be controlled (and is via our framework) to prevent a badly behaved optimization problem with many spurious local minima. Thus the global minimum of the relaxation is exactly the min cut.

Finally, we derive a connection between a special case of our framework and Laplace learning (proposition 4.1). This connection bridges our method with existing literature on Laplace learning. We consider guarantees about the generalization capability of our method to be an important future direction, but out of the scope of this work.

***Choice of $s$***
Several reviewers had questions regarding the choice of the $s$ parameter first introduced in equation 7 section 2.3. We highlight a result that appears in the main text and several experiments presented in the appendix of our paper. In particular, the choice of $s$ is governed by Proposition 3.2, which provides an upper bound on the choice of $s$ that guarantees recovery of integer-valued solutions (for intuition see figure 1). A trivial lower-bound is given by one convex relaxation of the problem ($s = 0$). Given the upper and lower-bound, we construct a homotopy path, a sequence of solutions $X(s)$ that vary in $s$. In practice, we choose a linear or geometric sequence of $s$ between $0$ and the upper bound. Many times, the upper bound need not be reached to recover a solution that is very close to integer (see Figure 6.b in the appendix). Likewise, we demonstrate that in practice, one can take a very coarse sequence of $s$. This result is presented and discussed in Figure 6 in appendix C.5. In fact, to reproduce the results in the main text, only a single nonzero data-dependent value of s is used (i.e. the homotopy path consists of constructing solutions for $s = 0$ and $s = 0.1$).

***Additional experiments***
In the table below we evaluate the effect mechanism used to construct the graph on MNIST. First, we evaluate two variants of weighting edges of the KNN-based method adopted in the main text. Singular refers to weights of the form $w_{ij} = 1/||x_i - x_j||$, where $x_i$ and $x_j$ are the features of the $i$-th and $j$-th data points. Uniform refers to $w_{ij} = 1$, i.e. an unweighted graph. In the third column, we evaluate a more sophisticated method for graph construction based on Non-Negative kernel regression (NNK) [1]. As opposed to the KNN-based methods, which only consider the distance (or similarity) between the query and the data, the NNK-based method takes into account the relative position of the neighbors themselves. Note that our CutSSL out-performs Poisson-MBO for the different graph weighting methods and NNK construction.
We thank the reviewers for this suggestion and we will add this experiment to the supplement in the final version of the paper.

| Method    | KNN (Gaussian) | KNN (Singular) |KNN (Uniform) |NNK |
| -------- | ------- |------- |------- |------- |
| Mean shift Laplace | 91.0 (4.7)    |  89.7 (2.8)  | 89.8 (2.3)    | 91.7 (1.1)    |
| PoissonMBO | 96.5 (2.6) | 94.2 (2.9) | 95.7 (1.6) | 97.3 (1.4)  |
| CutSSL    | 97.5 (0.1)    |  95.8 (2.6)    | 96.4 (2.0)    | 98.4 (1.5)    |

In the next experiment, we provide additional motivation for CutSSL. We show that in the absence of labels purely unsupervised graph cut-based methods (e.g. Spectral Clustering (SC)) yield a graph cut / partition such that the resulting clusters are well-aligned with the labels. To demonstrate this, we calculate a partition using both SC and a graph cut solution to equation 8 (‘cut’), i.e.this is CutSSL in the absence of labels. To evaluate the alignment of the clusters yielded by the cut with the labels, we solve a linear sum assignment problem using the true labels to find the best permutation of cluster assignments. This assignment of clusters to classes shows that  the graphs cut are relatively well-aligned with the class labels in contrast to Laplace learning (LP) which degenerates in the low label rate regime (i.e. just a single label per-class). Thus, cut-based techniques work since they are derived from a fundamentally non-degenerate framework and our CutSSL augments the multi-way graph partition with labeled nodes.

| Method    | MNIST | F-MNIST |  CIFAR10 |
| -------- | ------- |------- |------- |
| Laplace/LP |   16.1 (6.2)  | 18.4 (7.3)    | 10.4 (1.3)    |
| CutSSL    | 97.5 (0.1)   |  64.3 (4.6)   | 44.7 (5.9)   |
| Cut    |  95.9 (0.0)  |  62.7 (0.0)   | 41.1 (0.0)   |
| SC    | 92.6  (0.0) |   60.5 (0.0)  |  35.3 (0.0)  |


[1] Shekkizhar and Ortega, Neighborhood and Graph Constructions using Non-Negative Kernel Regression, ICASSP, 2020

---

### Decision · Program_Chairs · 2024-09-25

**Decision:**

Accept (poster)

**Comment:**

This paper proposes to formulate the graph-based semi-supervised learning problem as a continuous nonconvex quadratic program which exactly recovers integer solutions under proper conditions. An ADMM-type algorithm is developed for model optimization, with actual performance and scalability evaluated under a wide spectrum of label rates and data regimes on k-NN graphs. Also, the proposed formulation has been shown to bear similarities to the mean-shift Laplace learning heuristic and graph cuts.

The paper received four detailed reviews. Three reviews are in favor of accepting this work, while one review is inclined to reject it. After discussions, the reviewers reached a majority consensus that the proposed continuous partitioning strategy for graph SSL is novel and interesting, and the experimental results validates the practical usefulness of approach. One of the reviewers expressed a major concern regarding the discrepancy between motivation of study and theoretical justifications of the problem formulation in Eq. (9), which the authors did not deny and I agree indeed has room for imrovement. Also, math seems less rigurous in some places. For an instance, in Line 160, the matrix-vector product $Sm$ looks problematic as $S \in R^{M\times M}$ and $m\in R^{k}$ are generally incompatible in size.

Based on the reviews, authors responses and post-rebuttal discussions, it was assessed that this submission makes a sufficiently novel and solid contribution to the addressed problem. So the resulting sentiment is that the paper can be accepted given room in the program. The authors are encouraged to address the issues raised in the reviews as much as possible in the revised paper.